
# The lacustrine-water vapor isotope inventory experiment L-WAIVE

Patrick Chazette[1], Cyrille Flamant[2], Harald Sodemann[3], Julien Totems[1], Anne Monod[4], Elsa Dieudonné[5], Alexandre Baron[1], Andrew Seidl[3], Hans Christian Steen-Larsen[3], Pascal Doira[1], Amandine Durand[4], and Sylvain Ravier[4]

[1]Université Paris-Saclay, Laboratoire des Sciences du Climat et de l'Environnement (LSCE), CEA-CNRS-UVSQ, UMR CNRS 8212, CEA Saclay, 91191 Gif-sur-Yvette, France
[2]LATMOS/IPSL, Sorbonne Université, UVSQ, CNRS, Paris, France
Correspondence to: Patrick Chazette (patrick.chazette@lsce.ipsl.fr)
[3]University of Bergen, Bergen and Bjerknes Centre for Climate Research, Bergen, Norway
[4] Aix Marseille Univ, CNRS, LCE, Marseille, France
[5]Université du Littoral Côte d'Opale, Laboratoire de Physico-Chimie de l'Atmosphère (ULCO/LPCA), France

*Correspondence to*: Patrick Chazette (patrick.chazette@lsce.ipsl.fr)

**Abstract.** In order to gain understanding on the vertical structure of atmospheric water vapour above mountain lakes and to assess the respective influence of evaporation and advection processes, the L-WAIVE (Lacustrine-Water vApor Isotope inVentory Experiment) field campaign was conducted in the Annecy valley in the French Alps in June 2019. This campaign was based on a synergy between a suite of ground-based, boat-borne, and airborne measuring platforms implemented to characterise the thermodynamic and isotopic state above the lake environment using both in-situ and remote sensing instruments. Two ultra-light aircrafts (ULA), one equipped with a Rayleigh-Mie lidar, solar fluxmeters and an optical counter, and one equipped with a Cavity Ring-down Spectrometer (CRDS) and an in-cloud liquid water collector, were deployed to characterize the vertical distribution of the main stable water vapour isotopes ($H_2^{16}O$, $H_2^{18}O$ and $H^2H^{16}O$), and their potential interactions with clouds and aerosols. ULA flight patterns were repeated several times per day to capture the diurnal evolution as well as variability associated with different weather events. ULA flights were anchored to continuous water vapour and wind profiling of the lower troposphere performed by two dedicated ground-based lidars. Additional flights have been conducted to map the spatial variability of the water vapour isotope composition regarding the lake and surrounding topography. Throughout the campaign, ship-borne lake temperature profiles as well as liquid water samples at the air-water interface and at 2 m depth were made, supplemented on one occasion by atmospheric water vapour isotope measurements from the ship. The campaign period included a variety of weather events leading to contrasting humidity and cloud conditions, slope wind regimes and aerosol contents in the valley. The water vapor mixing ratio values in the valley atmospheric boundary layer were found to range from 3-4 g kg$^{-1}$ to more than 10 g kg$^{-1}$ and to be strongly influenced by the subsidence of higher altitude air masses as well as slope winds. A significant variability of the isotopic composition was observed within the first 3 km above ground level. The influence of the lake evaporation was mainly detected in the first 500 m of the atmosphere. Well-mixed conditions prevailed in the lower free troposphere, mainly above the mean altitude of the mountain tops surrounding the lake.

**Keywords:** L-WAIVE, valley, lake, Annecy, stable water isotopes, aerosol, lidar, airborne



## 1  Introduction

The vertical structure of the water vapor field in the lower troposphere is only sparsely documented in mountainous regions and particularly above Alpine mountain lakes (AMLs). This is in part due to the complexity and fast-evolving nature of the low-level atmospheric circulation in Alpine-type valleys which is intimately linked to the orography surrounding the lakes interacting with the synoptic scale circulation. Thermally driven wind systems may be induced by hilly terrain, such as slope, mountain, and plateau winds (Kottmeier et al., 2008). Such winds result in mountain venting phenomena that control the variability of the water vapour field in mountain catchment on very small-time scales. Furthermore, small-scale inhomogeneity in soil properties across a mountain valley, as well as lake breezes resulting from land-lake contrasts, may also induce the development of thermal circulations, particularly on clear-sky days, modifying the wind, humidity, and temperature fields on small spatial scales. The interaction of slope-driven and secondary circulations can furthermore influence the thermodynamical environment in the valley by the formation of convergence lines. Such convergence lines may favour the formation of shallow clouds, and in some cases even deep convection (Barthlott et al., 2006). Interaction with the synoptic scale circulation can lead to the formation of strong, gusty down-valley winds such as foehn events (Drobinski et al., 2007) and gap flows (Flamant et al., 2002; Mayr et al., 2007) that can also contribute to rapid modifications of the water vapour field in mountain catchment areas.

In light of the complexity around AMLs described above, two research questions arise: (1) What is the role of lakes on the local and regional atmospheric water cycle in AMLs? (ii) What is the relative contribution of evaporation from the lake to the atmospheric water content over and downwind of a lake? An early study based on water stable isotope measurements conducted in the US Great Lakes region suggested that in the summer up to 15% of the atmospheric water content in the atmosphere downwind of the lakes is derived from lake evaporation (Gat et al., 1994). Stable water isotopes have long been used as a tool to study processes in lacustrine and hydrologic systems (see review by Gat, 2010), as well as evapotranspiration (e.g. Berkelhammer et al., 2016). Isotopic measurements of lake water have shown the relative roles of evaporation from the lake surface and transpiration from surrounding vegetation (Jasechko et al., 2013). The link between hydrology and evaporation has mainly been investigated using vapour and liquid water isotopes measurements gathered just above the Earth's surface and samples from lake water and precipitation (e.g. Cui et al., 2016). Measurements from tall towers only provide incomplete information on the link between evaporation and atmospheric processes in the free troposphere, such as mixing and distillation (e.g. Steen-Larsen et al., 2013). He and Smith (1999) combined airborne measurements and surface sampling of the water isotope composition to study the evaporation process over the forests of New England, however before the advent of high-resolution laser-based spectrometers. More recent available airborne measurements of the isotope composition in the



boundary layer either focused on areas above sea (Sodemann et al., 2017), or did not include measurements of the surface isotope composition (Salmon et al., 2019).

Hence, unexploited potential remains to use stable water isotopes for increasing our understanding of the influence of evaporation, boundary-layer processes, and the free troposphere for local and regional climate conditions in AMLs. For example, the depth of the atmospheric layer over which the influence of evaporation from the lake surface is detectable, and how different factors control the depth of this layer are still largely unknown. Detailed and comprehensive analysis of small-scale factors, such as winds in a valley, and how they are related to the mesoscale and large-scale dynamics, such as synoptic scale subsidence in complex terrain, are therefore needed.

In order to get so insights into such aspects, the L-WAIVE (Lacustrine-Water vApor Isotope inVentory Experiment) field campaign was conducted in the Annecy valley (45°47' N, 6°12' E, in Haute-Savoie in the French Alps) around the Annecy lake during the month of June 2019. Being the second largest natural, glacial lake in France, the Annecy lake is expected to play a substantial role for the regional hydrometeorology.

The overarching scientific objective of L-WAIVE is to study evaporation processes and their heterogeneity over the Annecy lake using an original a multi-platform instrumental approach based on continuous high-resolution vertical profiling of tropospheric water vapour, temperature and wind as well as aerosols in the valley, together with ship-borne and airborne measurements of stable water isotopes ($H_2^{16}O$, $H^2HO$ and $H_2^{18}O$) in the lake, in the lower atmosphere as well as in-cloud and in precipitation. An additional objective is to construct a reference aerosol and water vapour stable isotope profiles database for ground, airborne, and satellite lidar simulators currently developed by the consortium of researchers involved in L-WAIVE, itself included in the WAVIL (Water Vapor and Isotope Lidar) project.

This paper provides an overview of L-WAIVE campaign in terms of experimental strategy (Section 2), instrumental platforms operations, environmental variables monitoring (Section 3) and synoptic conditions between 12 and 23 June 2019 in the Annecy lake area (Section 4). Atmospheric water vapour and liquid water isotopes, as well as lake water isotope observations made across the valley are described in Section 5. In Section 6, we summarize and conclude.

## 2    L-WAIVE experimental strategy

To achieve the scientific and methodological objectives of the L-WAIVE project, the field campaign was implemented in the southern part of the Annecy lake (the so called "Petit Lac"), in the vicinity of the city of Lathuile (Fig. 1). The Annecy lake is bordered by the city of Annecy to the north, the Massif des Bauges to the west (2217 m above mean sea level – a.m.s.l.), the Massif des Bornes to the East (2438 m a.m.s.l.) and the depression de Faverges to the south (where Lathuile is located). Lathuile is located east of the foothill of the Roc des Boeufs (1774 m a.m.s.l.) to the west of the "Petit Lac" and the Tournette summit (2350 m a.m.s.l.) to the east. The Annecy lake, at a mean altitude of 446.7 m a.m.s.l., covers an area of roughly 27.5 km$^2$, and has a mean (maximum) depth of 41.5 m (82 m).



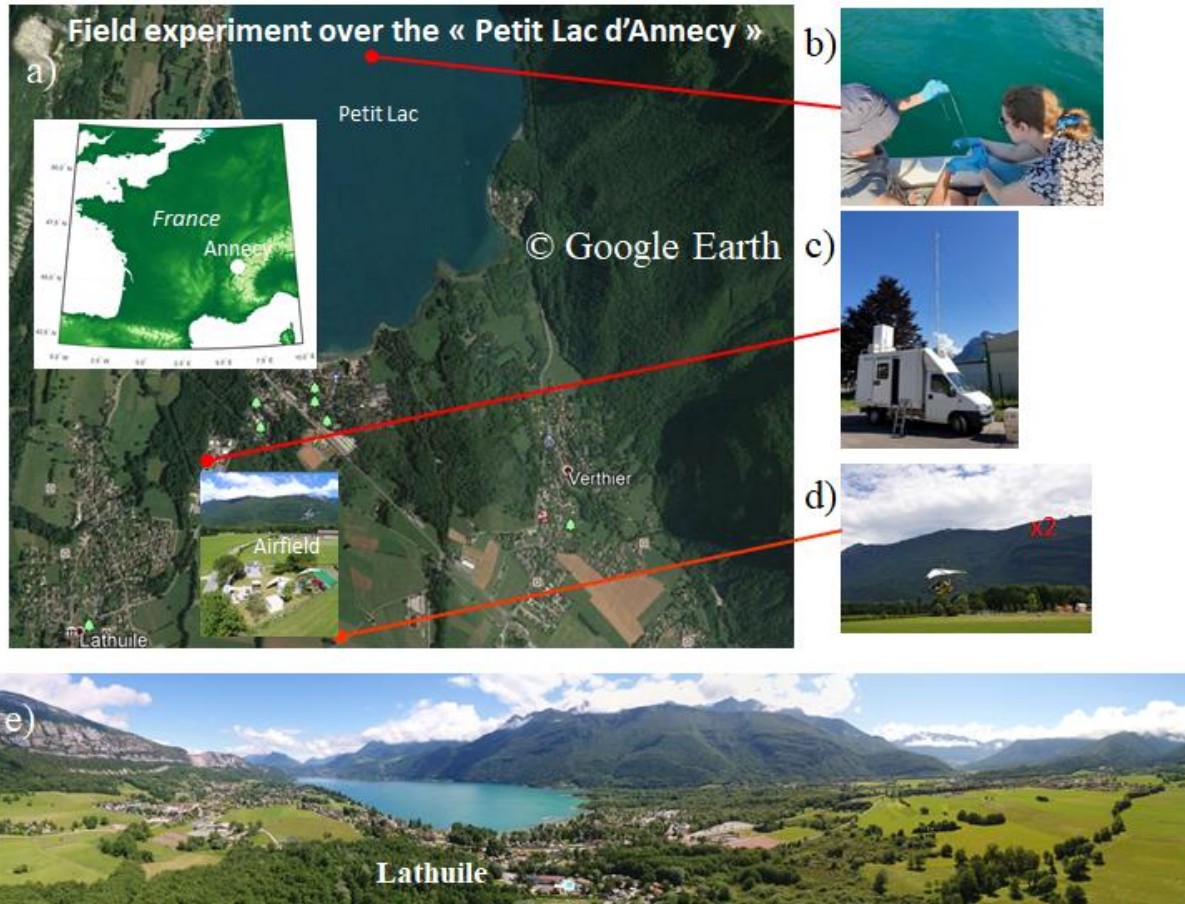

**Figure 1. Geographical location of L-WAIVE. The different pictures give a view of the environment where the measurements were performed and of the instrumented platforms used: a) location of the experiment, b) lake water sampling from a boat, c) instrumented van, d) instrumented ultra-light aircrafts, and e) panoramic view from UAV showing the location of Lathuile, as well as the Roc des Boeufs to the left, the Tournette summit to the right and the "Petit Lac" in between.**

### 2.1 Measurement platforms

Three airborne, one ship-borne and one ground-based instrumented platforms were deployed in the vicinity of Lathuile in order to monitor humidity, temperature, wind, clouds and aerosols in the lower troposphere over the Annecy lake and the surrounding valley environment, as well as to conduct measurements at the interface between the atmosphere and the lake, and in the lake. A brief overview of the platforms is given below, whereas the details on the instrumental payloads are given in Section 3:

    i)      Airborne platforms:

            a.     One ultra-light aircraft (ULA) was mainly dedicated to remote sensing measurements. It allowed exploring the two or three-dimensional structure of the lower troposphere thanks to a polarized Rayleigh-Mie lidar. It also carried a meteorological probe (pressure, temperature, relative humidity), an aerosol particle sizer and





two visible flux meters for the upward and downward solar radiations to complement the lidar measurements and determine the surface albedo in the visible spectrum. This will be referred to as "aerosol ULA" (ULA-A) in the following.

b. A second ULA carried both a Cavity Ring-down Spectrometer (CRDS) water vapour isotope analyser, a meteorological probe for pressure, air temperature, Global Positioning System (GPS) location and relative humidity, and a cloud water collector. The platform offers the opportunity to measure the vertical profiles of temperature, relative humidity, $H^2HO$, $H_2^{18}O$ and $H_2^{16}O$ and to collect cloud water samples. We will refer to this platform as "isotope and cloud ULA" (ULA-IC) in the following.

c. An Unmanned Aerial Vehicle (UAV) acquired vertical profiles of temperature, relative humidity, and pressure in the surface layer (first 150 m above the ground level) using a meteorological probe.

ii) Ground-based platform: Simultaneous high-resolution vertical profiles of water vapour, temperature, aerosols and winds were acquired continuously from two co-located ground-based lidars.

iii) Ship-borne platform: An instrumented boat allowed sampling the lake water by vials at the surface film and underneath (~2 m deep) for the assessment of $H^2HO$ and $H_2^{18}O$. A probe was also used to assess the vertical profiles of water temperature below the surface down to a depth of 55 m. A CRDS water vapour isotope analyser performed measurements during one day at the end of the experiment just above the lake surface in parallel with the lake water sampling.

These different platforms are presented in Fig. 1 together with a view of the experiment site.

## 2.2 Deployment

Depending on the weather conditions, airborne platforms were deployed several times a day to document the temporal evolution of the atmospheric boundary layer over the lake. The days of operation of all platforms are summarized in Table A1 of Appendix A. The ground-based water vapour, temperature and aerosol lidar operated continuously between 12 and 21 June in the morning (gathering over 220 h of data), while the ground-based wind lidar (WL) operated continuously between 14 and 23 June in the morning (acquiring also over 220 h of data). The UAV performed 7 flights between 13 and 21 June. A total of 22 successful flights were conducted with ULA-A between 13 and 19 June, while ULA-IC performed 15 flights between 14 and 20 June. The 15 ULA-IC flights were tightly coordinated with ULA-A flights. Valid cloud water samples were only obtained during the last 3 ULA-IC flights. The CRDS previously installed on ULA-IC was mounted on the boat from 21 to 22 June. Ship-borne CRDS observations were made at the surface of the lake on the last outings of the boat. Up to 12 lake water temperature profiles were made between 13 and 21 June (sometimes twice a day) while lake water samples were made on 14 occasions. Finally, 28 samples of precipitation were made during the campaign (7 of which on 15 June in the afternoon) between 11 and 22 June 2019. The distribution of the precipitation samples with time as seen in Table A1 gives an indication



as to when rain was encountered during the campaign, i.e. essentially at the beginning (14 and 15 June) and at the end (21 and 22 June) of the field deployment. In the meantime, clouds were regularly observed above the lake as documented by the ground-based lidars.

It is worth noting that all platforms (2 ground-based lidars, 2 ULA, 1 UAV, 1 boat) performed coordinated operations on 16, 17, 18, 19 and 20 June. These are considered as "golden days" and will be analysed in priority. A few less optimal days ("silver days") have also been defined based on the fact that 5 out 6 instruments were simultaneously in operations, namely: 13 and 21 June. 21 June stands out as being a day when both precipitation water samples, and ship borne CRDS were acquired and will also be an analysis priority.

On days when both ULA flew coordinated patterns (13, 16-20 June), flights typically began with a profiling sequence between the surface and ~4 km a.m.s.l. which was carried out in the vicinity of the 2 ground-based vertically pointing lidars (see Fig. 2). Soundings with levelled legs (see blue dotted line in Fig. 2a) were performed at a relatively slow ascent rate (~60 m min$^{-1}$) to ensure that the instruments were as close as possible to equilibrium with the environment. Upon reaching 4 km a.m.s.l., the flight route of the 2 ULA differed, ULA-A performing a high altitude survey above the Annecy lake (see red dotted line in

Fig. 2a), while ULA-IC was aiming for shallow cumulus clouds to sample cloud water droplets as illustrated in Fig. 2b showing the instrumental synergy that took place during L-WAIVE. Liquid water sampling was performed via multiple passes through the clouds to accumulate enough material to conduct isotope analysis. At the end of the flight, both ULA performed race-track descents around the ground-based lidars on their way back to the airfield.

During ascent and descent, the airborne lidar ALiAS onboard ULA-A was pointing sideways to directly derive the aerosol
extinction coefficient (Chazette et al., 2007; Chazette and Totems, 2017). For the exploration of the valley at a cruising altitude between 3.5 and 4.5 km a.m.s.l., ALiAS was pointing to the nadir. The combination of both flight sequences thus allowed to survey the 3-dimensional structure of the lower troposphere over the lake and its surroundings. The individual flight characteristics (time, maximum altitude, type of exploration) are presented in Appendix B for the two ULAs (Tables B1 and B2 for ULA-A and ULA-IC, respectively).





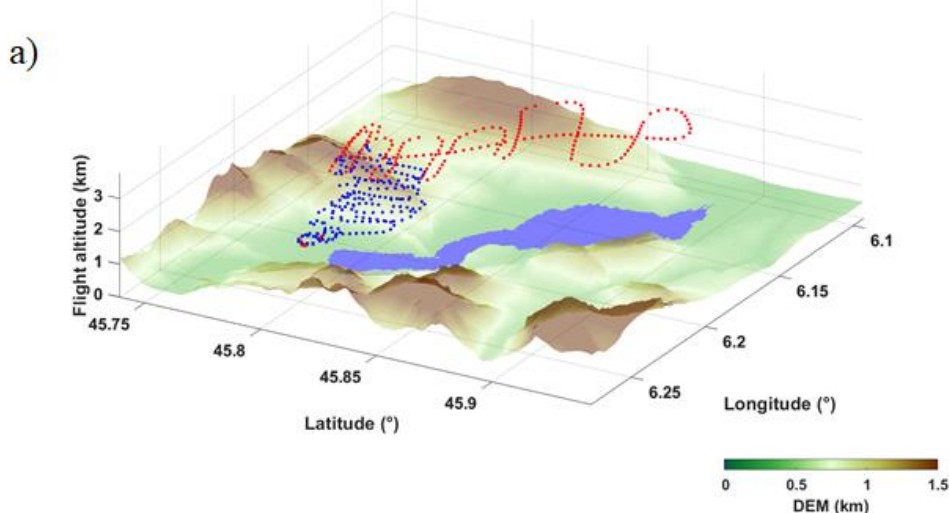

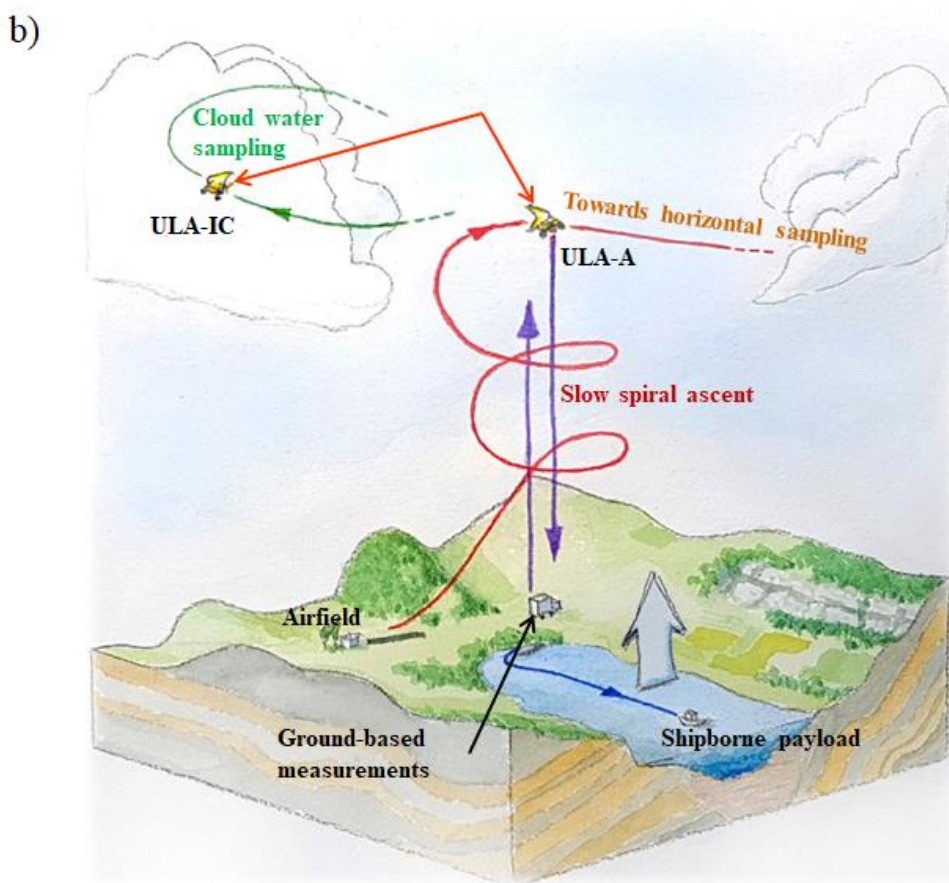


**Figure 2. Example of a typical ULA flight plan performed during L-WAIVE (performed on 17 June 2019. a) The flight track adopted during the flight (blue dots for vertical profiling and red dots for horizontal exploration above the lake). b) Schematic representation of the measurement strategy adopted during L-WAIVE.**



## 3 Instrumental set up on each platform

This section provides a detailed description of the payloads on all platforms deployed during L-WAIVE. The periods of operation of each measurement platform are given in Appendix A.

### 3.1 Airborne payloads

We used two Tanarg 912 XS ULA from the company Air Création (Chazette and Totems, 2017). For each ULA (ULA-A and ULA-IC, Fig. 3), the maximum total payload is of approximately 250 kg including the pilot. Flight durations were between ~1

and 2 hours, depending on flight conditions, with a cruise speed around 85-100 km h$^{-1}$. The ULA location was provided by a GPS and an Attitude and Heading Reference System, which are part of the MTi-G components sold by XSens.

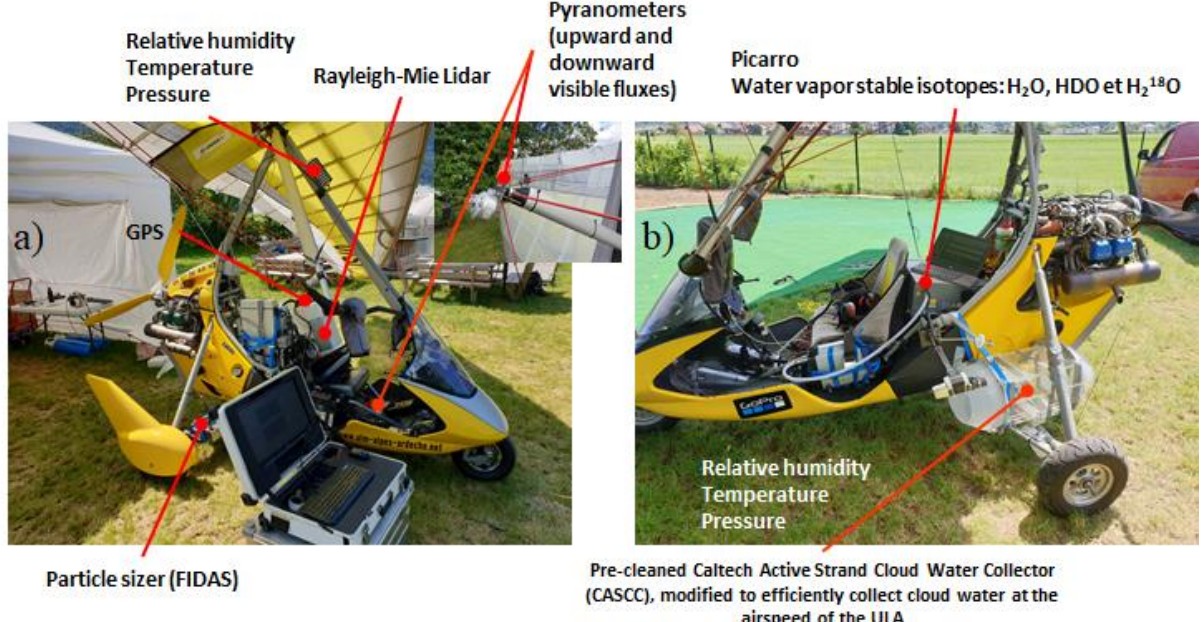

**Figure 3. Ultra-light aircraft payloads: a) remote sensing payload with the Rayleigh-Mie lidar ALiAS (ULA-A) and b) in situ payloads with the CRDS isotope analyser and the Caltech Active Strand Cloud Water Collector (ULA-IC).**

### 3.1.1 ULA-A

**Rayleigh-Mie lidar.** ALiAS was especially developed by LSCE as an airborne payload dedicated to aerosols and clouds samplings (Chazette et al., 2012). It emits a pulse energy of 30 mJ in the ultraviolet at 355 nm with a 20 Hz pulsed Nd:YAG laser (ULTRA) manufactured by QUANTEL™ (https://www.quantel-laser.com/). The acquisition system is based on a PXI (PCI eXtensions for Instrumentation) technology. The receiver contains two channels for the detection of the elastic backscatter

from the atmosphere in the parallel and perpendicular polarization planes relative to the linear polarization of the emitted radiation. The native resolution along the line-of-sight is 0.75 m, it is degraded to 15 m during the data treatment to improve





the signal to noise ratio. The wide field-of-view of ~2.3 mrad ensures a full-overlap of the transmit and receive paths close to 200-300 m from the emitter.

**Visible flux-meters.** Two pyranometers Kipp and Zonen™ CMP22 are mounted on ULA-A to determine the short-wave surface albedo. One is placed at the top of the wing's central mast to measure the downward flow and the other is placed under the cradle to measure the upward flow. The CMP22 pyranometer measures solar irradiance over the wavelengths from 200 to 3600 nm, with a directional error lower than 5 W m$^{-2}$.

**Meteorological probe.** Part of ULA-A payload was a shielded meteorological probe VAISALA PTU-300 for measuring temperature, pressure and relative humidity. This probe measures the atmospheric pressure, with a 1-minute sampling time, within an uncertainty of 0.25 hPa, the air temperature within an uncertainty of 0.2 K and relative humidity (RH) within a relative uncertainty of 2.5%.

**Particle sizer.** The granulometer used on board ULA-A was a FIDAS mobile manufactured by PALAS (https://www.palas.de/en/). The particle sizer operates on battery power with a volume flow of 1.4 l min$^{-1}$ in environmental conditions of temperature, atmospheric pressure, and relative humidity (no drying). The particle size distribution is determined from 180 nm to 20 µm by means of an optical aerosol spectrometer using Lorenz-Mie scattered light analysis. The LED source homogeneously illuminates an optically differentiated measurement volume with white light. Each particle moving through this volume generates a scattered light impulse detected at an angle of 85° to 95° degrees. The amplitude of the impulse is a measure of the particle diameter and the particle number corresponds the number of impulses. To allow in-flight measurement while limiting the loss of particles, a sampling head has been designed and printed in 3D in order to guarantee an isokinetic air flow at the entrance of the FIDAS.

### 3.1.2 ULA-IC

**CRDS water vapour isotope analyser.** ULA-IC carried a CRDS water vapour isotope analyser (L2130-i, Picarro Inc., Sunnyvale, USA; Ser. No. HIDS2254) for the in-situ measurement at about 5 Hz of the $H_2^{16}O$ mixing ratio, and the isotope ratios $\delta^{18}O$ and $\delta^2H$ for $H_2^{18}O$ and $H^2HO$, respectively. Water vapour was drawn into the spectrometer through an unheated inlet of 68 cm length (1/4 inch O.D. stainless steel with Silconert coating), pointing backward on the left side of the aircraft at a distance of 38 cm from the CRDS. Pointing forward next to the vapour inlet, a fast-response temperature and humidity probe (iMet XQ-2, InterMet systems, USA; Ser. No. 61124) measured thermodynamic properties (T, RH, p) and GPS location at 1 Hz. The CRDS analyser was operated in flight mode, with a flow of about 150 sccm through the inlet maintained by a membrane pump (Part No. S2003, Picarro Inc.). Pressure and water vapour mixing ratio were corrected using calibration functions established at the FARLAB laboratory, University of Bergen, Norway. Raw measurements of the isotope parameters, expressed as δ-values relative to VSMOW2 (Vienna Standard Mean Ocean Water 2, see Section 5), were corrected for the mixing ratio-isotope composition dependency using time-constant correction functions obtained by the method of Weng et al. (2020). For calibration, vapour isotope data were scaled onto the VSMOW2-SLAP2 scale by routine measurements of





secondary laboratory standards using a Picarro Standard Delivery Module (SDM, Part No. A0101, Picarro Inc.). Hereby,
secondary laboratory standards GSM1 ($\delta^2$H = -262.95±0.04‰, $\delta^{18}$O = -33.07±0.02‰), and DI ($\delta^2$H = -50.38±0.02‰, $\delta^{18}$O = -7.78±0.01‰) were measured repeatedly for 20 min each in the week prior to and after the campaign. Calibrations of the CRDS analyser in the laboratory before and after the field deployment showed minor drift during the measurement period. From long-term averages of the SDM calibrations, the combined uncertainty is estimated to be on the order of 0.5‰ for $\delta^{18}$O and 2‰ for $\delta^2$H. A small bubbler system was used for testing instrument drift before and after flight operations in the field. These tests
indicated an anomaly in the CRDS measurements 18-20 June, partially affecting two flights in the morning of 19 June, and the first flight on 20 June. The anomaly was due to a saturated inlet system from condensate forming on the aircraft during a cloud sampling flight on 18 June. Flight periods affected by the saturated inlet were excluded from further analysis. Further details of the data processing and calibration procedure are described in the data report that accompanies the data set, and in a forthcoming publication.

In the afternoon of 19 June 2019, the two ULA performed a coordinated ascent, providing an intercomparison of response times and measurement offsets regarding pressure, temperature, and relative humidity on the two aircraft (Fig. 4). During the entire ascent sequence, there is a small, visually discernible temperature offset between the two instrumental packages (Fig. 4b). This offset originates from the slower response time of ULA-A instrumental package (red line), lagging behind ULA-IC the faster and more exposed iMet probe (black line). The relative humidity measurement on ULA-IC is clearly faster,
and resolves spikes in more detail (Fig. 4c, black line). A comparison of the specific humidity calculated from iMet and the laser spectrometer revealed similar response times (not shown). The intercomparison between the two ULAs thus also provides a first-order in-flight validation of the water vapour measurements of the CRDS analyser.

A time resolution better than 0.1 Hz was obtained for specific humidity from the CRDS and iMet probe, providing a spatial resolution of 200-300 m in horizontal direction, and 10-50 m in the vertical, assuming a typical horizontal speed of 80-100 m
s$^{-1}$ and ascent rate of the aircraft of about 1-5 m s$^{-1}$. Due to more complex memory effects, the isotope composition has lower actual time resolution (Sodemann et al., 2017; Steen-Larsen et al., 2014). We use here 10 s average data for all parameters on ULA-IC from upward profiles, filtered for rapid elevation changes, defined as exceeding 50 m ascend or 20 m descend within 10 s.





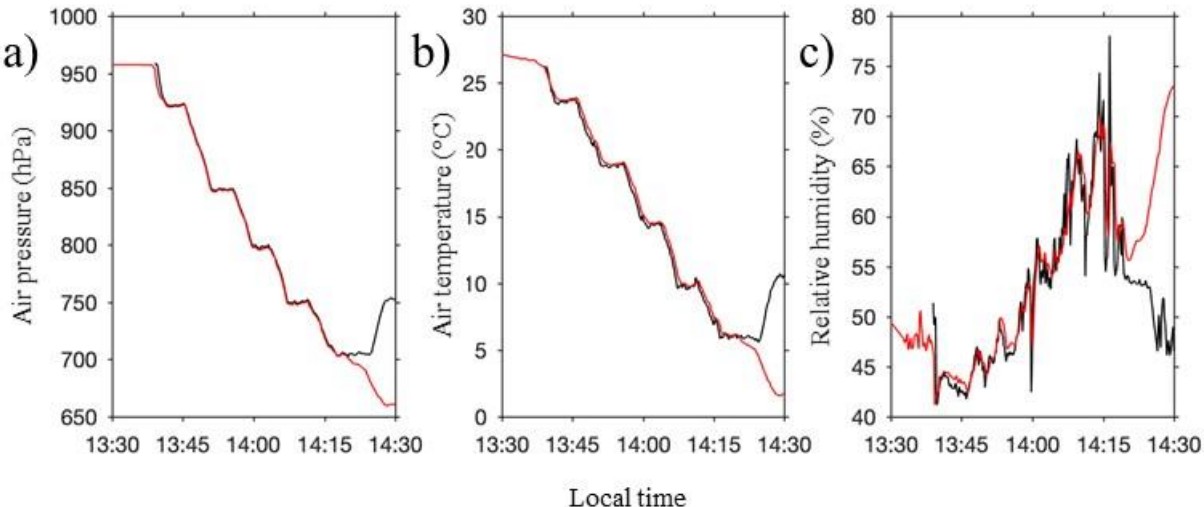

Local time

**Figure 4. Comparison between (a) air pressure (hPa), (b) air temperature (°C) and (c) relative humidity (%) between the iMet probes mounted on ULA-A (red line) and ULA-IC (black line) during coordinated flights in the afternoon of 19 June 2019.**

**Cloud water collector.** A pre-cleaned Caltech Active Strand Cloud Water Collector was mounted on ULA-IC, modified to efficiently collect cloud water at the speed of the ULA (Fig. 3b). The ULA-IC relative cruising speed is 85 to 100 km h$^{-1}$, in the operating range of the Caltech Active Strand Cloud Water Collectors (CASCC, Demoz et al., 1996). A CASCC was modified to efficiently collect cloud liquid droplets from the ULA. In order to sample droplets under the same conditions as those obtained on the ground, the CASCC's fan was removed, and its inlet and outlet were prolonged with convergent and divergent High Density PolyEthylene (HDPE) cones to ensure an isokinetic air sampling. The flow through the probe must be steady and as turbulence-free as possible. The design of these modified inlet and outlet was calculated to get a constant mass of water droplets per time unit through the probe onboard the ULA. All sampling materials used were plastic or HDPE. The resulting probe was installed on the side of the ULA where the flow was assumed laminar and allowed air sampling with a flow rate of 35 to 47 m$^3$ min$^{-1}$, ahead of the motor exhaust (Fig. 3b). The CASCC strings and inlet were pre-cleaned with deionized water prior each flight and covered with a clean plastic bag when not in-cloud (especially during take-off and landing). For a flight of 10 minutes inside a shallow cumulus cloud, the probe typically collected 41 to 48 g of cloud liquid droplets corresponding to a liquid water content of 0.10 to 0.16 g m$^{-3}$, typical for such cloud type (Herrmann, 2003).

**3.2     Ground-based instrumentation**

The ground-based scientific facility hosted by the technical department of the city of Lathuile was mainly composed of the Raman lidar WALI and of a scanning Doppler lidar. WALI was embedded in the ground-based MAS (mobile atmospheric station; Raut and Chazette, 2009). The UAV was also operated from this site, close to the lidar. Figure 5 shows the two lidars and the UAV.





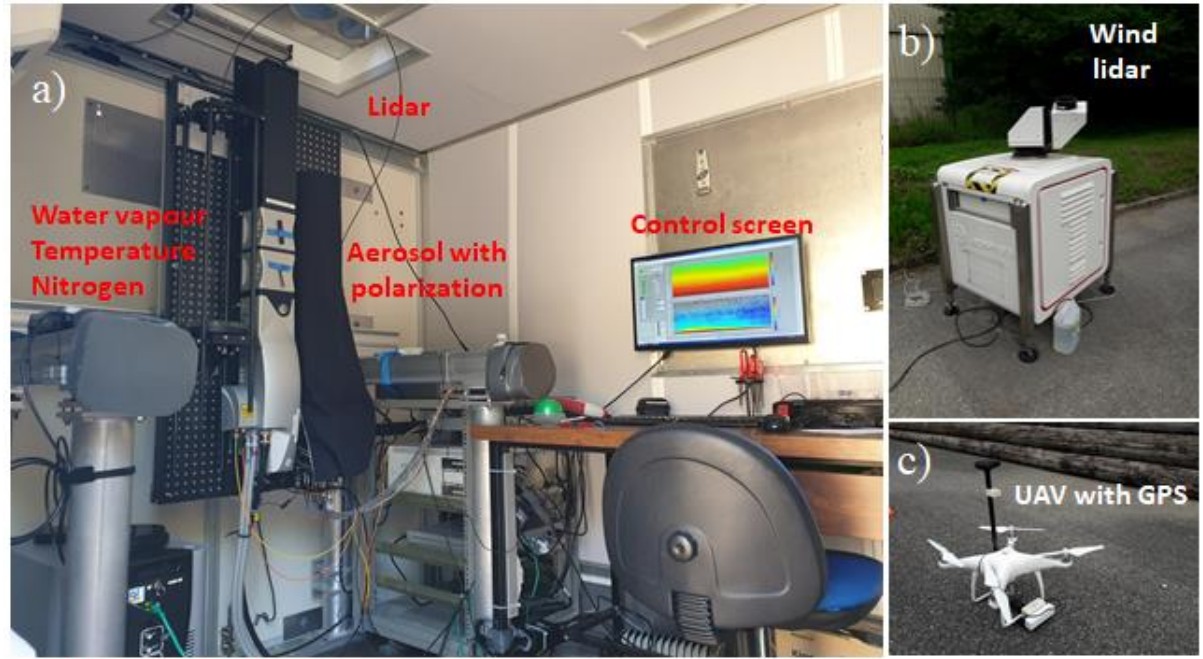

**Figure 5. a) Lidar WALI on board the mobile atmospheric station (MAS), b) wind lidar located close to the MAS, and c) UAV with its GPS antenna (in black).**

### 3.2.1 Raman lidar

WALI has been developed at LSCE (Chazette et al., 2014) based on the same technology as its precursor instruments LESAA (Lidar pour l'Etude et le Suivi de l'Aérosol Atmosphérique; Chazette et al., 2005) and LAUVA (Lidar Aérosol UltraViolet Aéroporté, Chazette et al., 2007; Lesouëf et al., 2013; Raut and Chazette, 2009). It is a custom-made instrument dedicated to atmospheric research activities.

The lidar operates with an emitted wavelength of 354.7 nm and is designed to fulfil eye-safety standards (EN 60825-1). Its emitter is a pulsed Nd:YAG laser (Q-smart 450 by Quantel™) with a fibre laser injector to stabilize the emitted wavelength. The acquisition system is based on a PXI (PCI eXtensions for Instrumentation) technology manufactured by the National Instruments™ company, and contains 12 bits digitizers at 200 MS/s corresponding to a native vertical resolution of 0.75 m. During the entire experiment, the acquisition was performed for mean profiles of 1000 laser shots leading to a native temporal sampling close to 1 minute. The UV pulse energy was ~70 mJ and the pulse repetition frequency was 20 Hz. The wide field-of-view of ~2.3 mrad allows a full-overlap of the transmission and reception paths beyond ~ 200-300 m. Note that the pulse energy decreased linearly with time due to a laser failure, from 70 mJ to about ~50 mJ at the end of the campaign.

The receiver is composed of 2 distinct detection paths, both using small collector diameters of 15 cm, with a low full-overlap distance (~150-200 m). The first path is dedicated to the detection of the elastic molecular, aerosols and cloud backscatter from





the atmosphere. Two different channels are implemented on that path to detect i) the total (co-polarized and cross polarized with respect to the laser emission) and ii) the cross-polarized backscatter coefficients of the atmosphere. The second path, a
fibered achromatic reflector, is dedicated to the measurements of the atmospheric Raman scattering, namely the vibrational signal for nitrogen ($N_2$-channel) and water vapor ($H_2O$-channel) and the rotational signal to derive the temperature (T-Channel).

The water vapor mixing ratio (WVMR) is retrieved with an absolute error less than 0.4 g kg$^{-1}$ in the first 2 km above the ground level (a.g.l.)  (Chazette et al., 2014; Totems et al., 2019). The calibration of the T-channel is derived from the methodology
presented by Behrendt (2006) and leads to an absolute error on the temperature lower than 1 °C within the first 2 km a.g.l. The final vertical resolution is set to 15 m below 1 km a.g.l. and 30 m above, and the temporal resolution is 0.5 h. In the following, a temporal resolution of 1 h is used.

### 3.2.2    Wind lidar

Wind profiles were measured using a scanning Doppler lidar (Leosphere Windcube WLS100). It operates in the infrared (1.543
μm) with a low pulse energy (0.25 mJ) but a high pulse repetition frequency (20 kHz). The Doppler shift due to the particles' motion along the beam direction (radial wind speed) is determined through heterodyne detection followed by fast Fourier transform analysis. The acquisition time was set to 1 s during the campaign. The pulse duration is 200 ns, corresponding to an axial resolution of 50 m given the pulse shape, with a minimal and maximal range of 100 m and 7.2 km, respectively. The axial resolution can be lowered to 25 m by reducing the pulse duration (100 ns) while increasing the pulse repetition frequency
(40 kHz) which in turn reduces the minimal and maximal range to 50 m and 3.3 km, respectively. In practice, the maximum range is limited by the signal level. A minimum Carrier to Noise Ratio (i.e. signal to background noise ratio) of −27 dB is required to keep the radial wind uncertainty (determined from the spectrum peak width) below 0.5 ms$^{-1}$. Therefore, observations in the free troposphere are possible only when elevated layers of aerosols are present. Even in the boundary layer, several days of very low aerosol load occurred during the campaign, in which case the Carrier to Noise Ratio threshold was
lowered to −30 dB. With such a low Carrier to Noise Ratio, the measurements must be considered with caution.

Profiles of the three components of the wind vector were determined using the Doppler Beam Swinging technique originally proposed for Doppler radar (e.g. Koscielny et al., 1984). Here, the measurement cycle includes one vertical beam, for which the radial wind is the vertical component of the wind vector, and four slanted beams (15° from zenith) in the cardinal directions to derive the two horizontal components of the wind. During L-WAIVE, the lidar alternated between 20 cycles of 25 m-
resolution profiles and 20 cycles of 50 m-resolution profiles. The uncertainty on the horizontal / vertical wind components is determined using the variability inside averaging periods, possibly gathering several measurement cycles. Regarding the effects of orography, corrections to the wind profiles are needed when working over sloping terrain or on top an elevated area (Bradley et al., 2015). During the campaign, the Doppler lidar was positioned in a wide valley (~3 km at the observation site)



with a rather flat bottom, and the closest distance existing between one of the slanted beams and an obstacle was ~1 km. Therefore, the influence of the orography on the measurement are assumed to be negligible here.

### 3.3 Shipboard payload

#### 3.3.1 Lake and atmospheric sampling

The shipboard payload is shown in Fig. 6. The lake water surface and subsurface were sampled at the middle of the "Petite Lac" to measure water isotopes and chemicals from the boat.

The lake water thermal stratification was monitored using an EXO sonde, equipped with temperature, pressure, pH, dissolved dioxygen, ion conductivity and chlorophyll sensors. Profiles recorded in the middle of the "Petit Lac" (Fig. 1) once or twice per day (Table A1) and showed that the thermocline (Fig. 7) was below a depth of 10 m, in good agreement with previous studies of the Lake (Danis et al., 2003).

Subsurface water samples were collected at a depth of 2 m in HDPE capped flasks. Surface water samples were collected using
a 30 x 30 cm silica glass plate immersed into water for a minute, then gently removed from water vertically (Cunliffe et al., 2013). The water falling from the plate in a continuous flow was not sampled, then, the dropwise water was collected in HDPE capped flaks. The surface microlayer samples were then collected by scraping the remaining water on the glass plates (using a rubber scraper) in amber glassware capped vials. On 22 June, the CRDS isotope analyser taken on board the boat to sample a cross-section of the "Petit Lac" through the location used for in situ sampling on the other days. This made it possible to
follow the evolution of isotope concentrations as a function of the distance from the shore and the depth of the lake on that day. Water vapour isotope measurements were taken from an inlet at either ~20 cm or ~2 m above the water surface, while the iMet sonde measured temperature, relative humidity and location. Post-processing and calibration of water vapour measurements were done as for the aircraft data (Sec. 3.1.2).

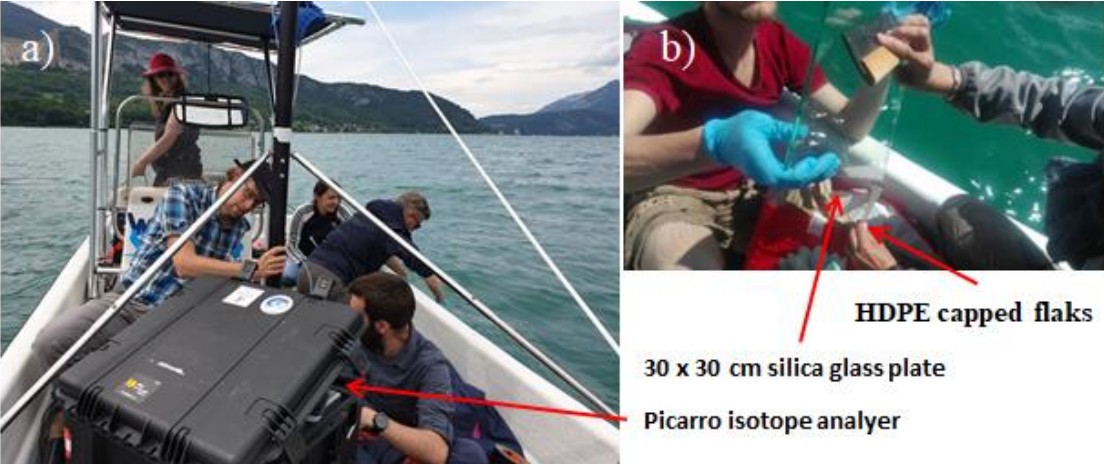

**Figure 6. Illustration of the shipboard payload with operators. a) The CRDS isotope analyser is on board the boat. b) The surface microlayer of the lake is sampled using both a silica glass plate and a high-density polyethylene (HDPE) capped flask.**





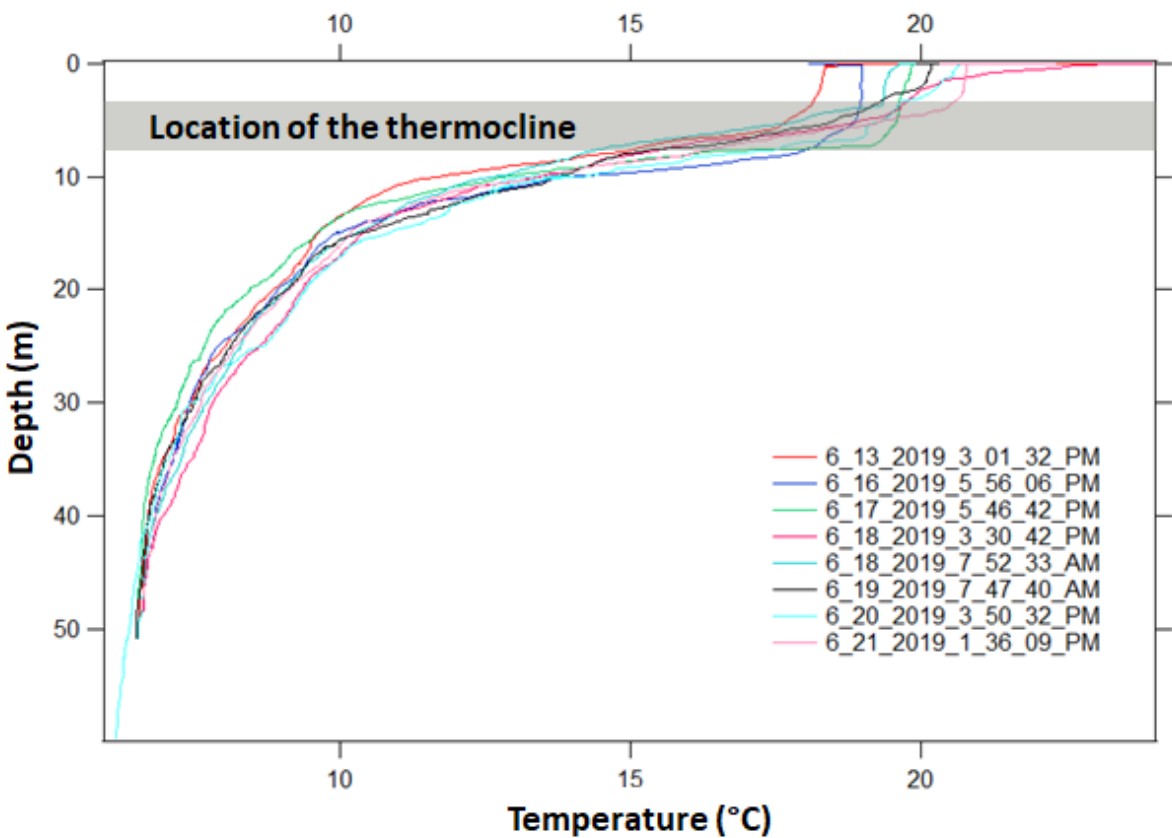

**Figure 7. Vertical temperature profiles in lake water for different days. The grey area highlights the location of the thermocline.**

### 3.3.2 Isotopic analysis of liquid water samples

Liquid samples were measured for isotopic composition ($H_2^{18}O$, $H^2H^{16}O$) at FARLAB, University of Bergen according to standard laboratory procedures. For isotopic analysis, samples were transferred to 1.5 ml glass vials with rubber/PTFE septa (part #548-0907, VWR, USA). An autosampler (A0325, Picarro Inc) injected ca. 2 µl per injection into a high precision vaporizer (A0211, Picarro Inc, USA) heated to 110°C. After blending with dry $N_2$ (< 5 ppm $H_2O$) the gas mixture was directed into the measurement cavity of a Cavity-Ring Down Spectrometer (L2140-i, Picarro Inc, Sunnyvale, USA) for about 7 min

with a typical water concentration of 20 000 ppm.

Three secondary laboratory standards were measured at the beginning and end of each batch for calibration purposes. Batches consisted typically of 20 samples, with laboratory drift standard DI, measured every 5 samples. For calibration according to IAEA recommendations, 16 injections of the laboratory standards EVAP ($\delta^2H = 4.75 \pm 0.11$‰, $\delta^{18}O = 5.03 \pm 0.02$‰) and GSM1, were used, and averaged over the beginning and end of each batch for calibration. Long-term measurement precision is 0.66‰



for $\delta^2H$, 0.15‰ for $\delta^{18}O$, and 1.05‰ for the d-excess, evaluated from the 1-sigma standard deviation of the analysis of an internal laboratory standard over one year.

## 4     Meteorological conditions during L-WAIVE

### 4.1     Synoptic conditions

During L-WAIVE, France was under the influence of two main synoptic features, namely a pronounced trough over Britany
and the British Isles and a high pressure ridge extending from north Africa across the Mediterranean and all the way into eastern Europe. This weather situation is illustrated in the Fig. 8 using the Fifth European Centre for Medium-Range Weather Forecasts Reanalysis (ERA5). This configuration caused a particularly strong pressure gradient over the western Mediterranean for the period 12-16 June, flanked in the east by an intense Libyan anticyclone (Fig. 8a). The ridge weakened and broadened over the following days (17-20 June, Fig. 8b) before strengthening again (21-23 June) as the Libyan high intensified (not
shown).

The synoptic flow at 700 hPa over the Annecy lake was dominantly south-westerly from 12 to 16 June, except on 14 June when it was almost southerly as the western flank of the high pressure ridge was seen to straddle the Alps (Fig. 8a). On that day, Saharan dust passed over the experimental supersite (not shown). Between 16 and 17 June the synoptic flow over the Alps was substantially weaker, with jets located north and south of central Europe (Fig. 8b) quite unlike the previous period.
Between 18 and 23 June, the synoptic south-westerly flow over the area of interest intensified again as the high-pressure ridge built up.

During the course of the campaign, the area of interest was alternatively under the influence of warmer temperatures (air masses with high equivalent potential temperature) linked with the high pressure ridge (14-15, 19-23 June, Fig. 9a) and colder temperature (air masses with low equivalent potential temperature) associated with the surface low over the British Isles (12-
13, 16-18 June, Fig. 9b).



**Figure 8. ECMWF ERA5 geopotential height at 700 hPa and horizontal winds at 700 hPa (black vectors) on (a) 14 June and (b) 17 June 2019. The location of Lathuile is indicated with a red dot surrounded by white.**



**Figure 9. Same as Fig. 8, but for equivalent potential temperature at 700 hPa.**

Due to the varying intensity of the high-pressure ridge, the direction of the mean flow at 925 hPa in the Rhone Valley and nearby Annecy lake varied substantially from one day to the next between 16 and 19 June. For instance, a well-established mistral flow was blowing in the Rhone Valley on 16-17 June (Fig. 10a) while strong southerly flow was observed on 18-19 June (Fig. 10b). Before 16 June and after 19 June, the 925 hPa winds in the area of interest were weak and had no dominant direction. According to ERA5 reanalysis, mostly dry condition prevailed over the Annecy lake (RH below 60%), except on 14-15 and 17 June when RH were more than 80%. It is worth noting that 15 June is the day when most precipitation samples


were gathered during the campaign (8 out of the total 28, Table A1). Several isolated thunderstorms were observed on that day, and on 16 June in the morning. The influence of weather events during the campaign will be further discussed in the following sections.


**Figure 10. ECMWF ERA5 relative humidity at 925 hPa (color) and horizontal winds at 925 hPa (black vectors) on (a) 16 June and (b) 18 June 2019. The location of Lathuile is indicated with a red dot surrounded by white.**

**4.2     Scattering layers as tracers of atmospheric dynamics**





Rayleigh-Mie lidars are very efficient tools to detect aerosol layers, but also clouds (e.g. Platt, 1977; Berthier et al., 2004),
both semi-transparent (ice clouds) or opaque (liquid water clouds) to the laser beam. During L-WAIVE, the temporal evolution
of the aerosol burden and clouds above the Annecy lake have been monitored using both the ground-based lidar WALI and
the airborne lidar ALiAS. In addition, continuous monitoring of aerosols vertical distribution from a lidar allows to get insight
into air masses transport. The identification of their particulate constituents (e.g. dust, pollution aerosols,…) may also provide
information on the origin of observed air masses. For instance, such a capability was used to improve our understanding of
atmospheric circulation in complex situations such as extreme heat wave phenomena (Chazette et al., 2017).

Fig. 11 shows the temporal evolution of the vertical profile of the aerosol scattering ratio (ASR, Fig. 11a) and of the linear
volume depolarization ratio (VDR, Fig. 11b), as defined in Appendix C. It can be seen that the vertical extension of aerosols
in the valley is highly variable over time as is the presence and nature of clouds during the course of the campaign. There is a
strong diurnal cycle of ASR in the valley which is probably related to the slope winds. The depth of the aerosol layer is largest
around 0000 LT when downslope winds tend to favour the accumulation of aerosols in the valley, while it is exhibits a
minimum around 1200 LT when upslope wind tend to flush aerosols out of the valley. Moreover, significant day-to-day
variations of the aerosol layer depth are observed with the lidar-derived ASR which highlights local, regional and even synoptic
scale transport of air masses over the Annecy valley. At the beginning of the campaign (mainly on 14 June) large VDR values
are detected up to altitudes exceeding 7 km a.m.s.l. (Fig. 11b). These large VDR values are related to a large-scale Saharan
dust transport episode over the Lac d'Annecy valley favoured by the synoptic conditions on that day (see Section 4). These
aerosols are progressively mixed downward by subsidence, reaching the valley floor around 0000 LT on 15 June. Slightly
enhanced VDR values are also evident on 19 and 20 June that are associated with local and/or regional pollution advected in
altitude and verticaly mixed by dry convection.

The period was rather cloudy as shown in Fig. 11a (shaded areas) where cloud types are indicated as the raining periods
including thunderstorms. It is worth noting large lidar-derived VDR values detected in the presence of dense clouds are related
to multiple scattering (areas appearing in pink in Fig. 11b), not to the presence of non-spherical aerosols.

The measurements made with the airborne particle sizer corroborate the nature of the particles identified from the lidar
observations. Figure 12 shows that on 14 June, larger particles are observed above 2.5 km a.m.s.l., compatible with the presence
of dust-like particles in the lower troposphere. It should be noted that on 19 and 20 June, the mean radius of the particles does
not exceed 0.4 µm, suggesting the presence of pollution aerosols mixed with some industrial dust-like particles explaining
anhanced VDR values. The altitude distribution of particle size on 19 and 20 June are also quite different, with large
concentrations of small particles being observed from the surface to 4 km a.m.s.l. on 19 June indicating more efficient dry
convection, whereas on 20 June, the largest particle concentrations are only observed below 2 km a.m.s.l.




Figure 11. Temporal evolution of a) the aerosol scattering ratio (ASR) where the shaded areas correspond to the presence of clouds and b) the linear volume depolarization ratio (VDR) from 13 to 21 June 2019. The cloud type and rain location are indicated, as is the type of the main aerosol structures identified by the VDR (dust or pollution aerosols). Cs, Ns, As, Ac and Cu indicate cirrus, nimbostratus, altostratus, altocumulus and cumulus clouds, respectively.





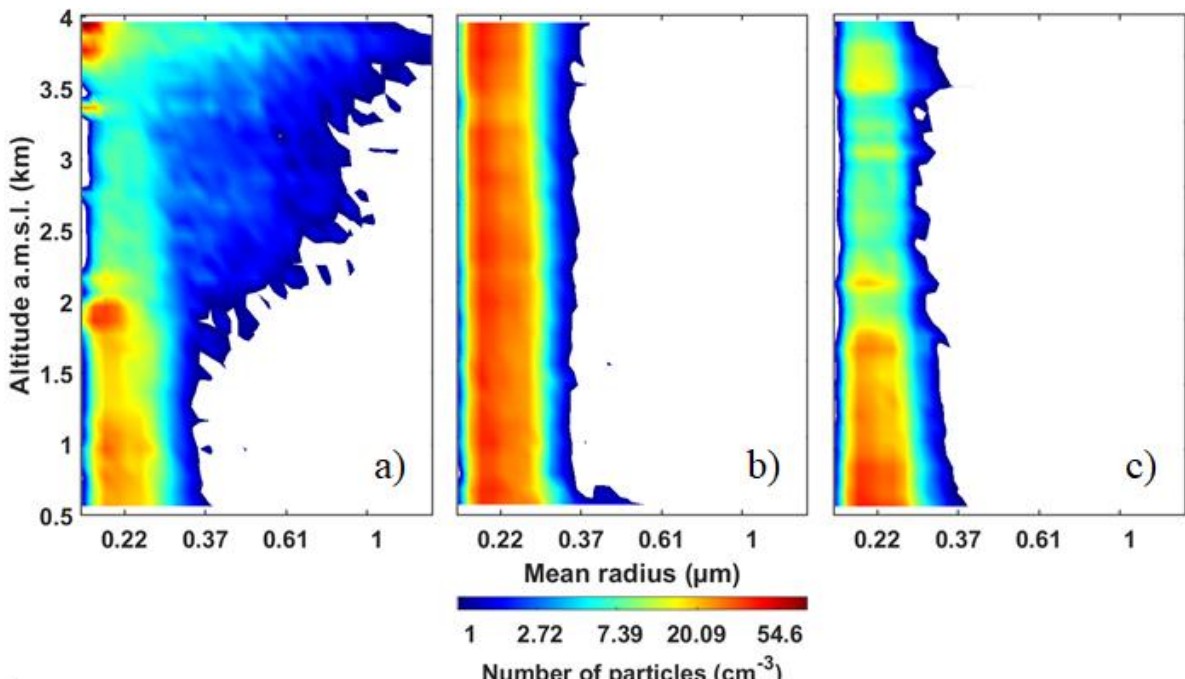

**Figure 12. FIDAS-derived aerosol size distribution for the flights a) F05 on 14 June (long-range transport), b) F17 on 19 June (local pollution) and c) F18 on 20 June 2019 (regional pollution).**

### 4.3 Local wind

The temporal changes in wind intensity and direction observed by wind lidar are given in Fig. 13. Weak winds, generally less than 5 ms[-1], are observed in the valley, below 2 km a.m.s.l. The wind intensity does not show repetitive patterns from one day to the next. In contrast, wind direction shows regular variations, with winds directed towards the exit of the valley (south) during the day, consistent with the flushing of aerosols suggested by the lidar observations. During the night, winds are directed towards the city of Annecy, thereby contributing to the accumulation of aerosols in the Annecy lake catchment. The local wind in the valley is obviously disturbed during stormy periods and during the episode of dust long-range transport. On 14 June 2019, the presence of these aerosols at higher altitudes (up to ~5 km a.m.s.l.) allowed the wind to be retrieved with the WL above 2 km a.m.s.l. Strong southerly winds (in excess of 20 m s[-1] above 3.5 km a.m.s.l.) were observed in agreement with the meteorological fields in Fig. 8a. The upper part of the regional pollution plume on 20 June is also seen to be associated with stronger southwesterly winds in excess of 10 m s[-1] (Fig. 13a).





**Figure 13. Temporal evolution of a) wind speed (m s⁻¹) and b) wind direction (0° is for North and 90° for east) obtained from wind lidar.**


## 5    Water vapor in the low troposphere during L-WAIVE

Water in vapor phase was sampled throughout the campaign in terms of mixing ratio of the main isotope and abundance of isotopes $H_2^{18}O$ and $H^2H^{16}O$. At the same time, liquid water samples were taken from the surface of the lake and in the clouds to extract their isotopic contents to place them in the context of the atmospheric water vapor of the lower troposphere.





### 5.1    Ground-based lidar measurements and analysis


The vertical profiles of WVMR derived from the WALI ground-based lidar are shown in Fig. 14a. They show highly variable water vapor in the first 2 kilometers of the atmosphere. Lower WVMR values are associated with nighttime downslope winds for instance, and higher values are generally associated with upslope valley winds but may persist during days associated with heavy precipitation such as on 15 June 2019. When northerly winds prevail in the planetary boundary layer of the valley, air


masses are advected along the lake axis before reaching the measurement site, leading to an increase of water vapor associated with the evaporation of the surface water of the lake. The influence of the lake is thus seen mainly up to altitudes between ~1 and 2 km a.m.s.l. during the day and significantly lower the rest of the time ($\lesssim$ 0.5 km a.m.s.l.).

Some of the marked WVMR features observed in the lidar measurements are reproduced in the WVMR fields of ERA5 with an horizontal resolution of 0.25° (Fig. 14b), especially during the second part of the campaign, starting on 18 June 2019. They


differ more in the first part which is influenced by thunderstorms and strong rainy periods, and thus more local processes. The boundaries between the different moist air mass types are similar between lidar measurements and reanalyzes, although the reanalyzes are moister in the planetary boundary layer. There is a strong decrease of water vapor at ~3 km a.m.s.l. which marks the transition to the free troposphere where long-range air mass transport occurs. This altitude corresponds roughly to the average altitude of the mountains at the regional scale around the measurement site.


The WVMR values retrieved from the meteorological probe on board the 2 ULAs match well with those derived from the lidar, as shown in Fig. 15 for the ULA flights listed in Table B1 and B2. Furthermore, the in-situ measurements made with the 2 meteorological probes are in excellent agreement with one another. A drier lower troposphere is indeed found during the first two flights on 23 June and the transition to the part of the troposphere influenced by long-distance transport between ~2.5 and 4 km a.m.s.l. is well represented. Differences in the altitude of the transition layer are seen between the in-situ and lidar


observations which are related to the fact that the airborne measurements were sometimes taken over the entire valley and even over the surrounding mountains.





**Figure 14: Temporal evolution of the water vapor mixing ratio (WVMR) derived from the ground-based Raman lidar and b) ERA5 reanalysis at 0.25° horizontal resolution from 14 to 21 June 2019 over Lathuile. White areas in a) correspond to missing data during the daytime above 2-3 km a.m.s.l. caused by detection limitations of the lidar.**






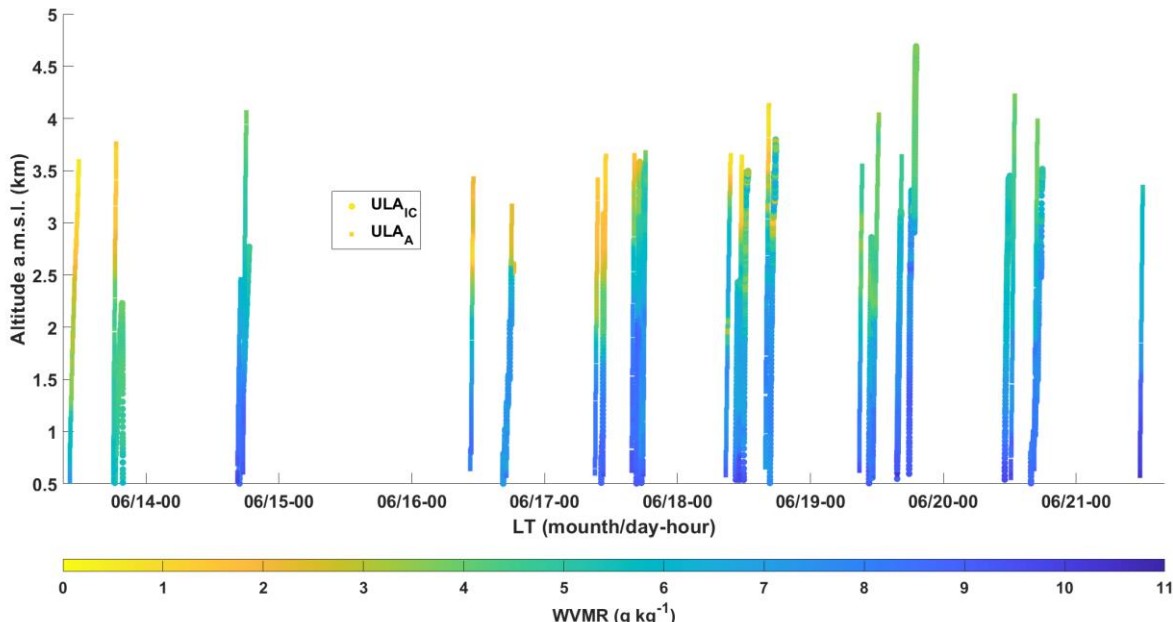

**Figure 15: Vertical profiles of the water vapor mixing ratio (WVMR) derived from the meteorological probe on board the two ULAs.**

## 5.2  Water isotope survey of the lake d'Annecy region

Stable isotope content of water vapour and in liquid water was quantified regarding $H_2^{18}O$ and $H^2H^{16}O$, as well as the deuterium
excess (d-excess*)*. Water vapour was sampled in-situ using a CRDS, whereas liquid samples, including precipitation, cloud
water and lake water were analysed in the laboratory. Here we report the isotope composition (see interpretative framework in
Appendix D) as delta-values relative to a standard (e.g. Gat, 1996).

### 5.2.1  Atmospheric water vapor sampling

In total, 15 flights with ULA-IC have been performed (see Table B2) including 14 flights where the CRDS allowed a
representative sampling of $\delta^2H$ and $\delta^{18}O$ (saturation on the inlet was encountered throughout flight 11). We provide here an
overview of the acquired measurement data, while a more detailed analysis on this dataset is ongoing. The isotope content
$\delta^2H$ observed during the flights ranged between about -340 and -80‰ for flight altitudes up to ~3.5 km a.m.s.l. as shown in
Fig. 16a. For this figure, we considered only the ascending parts of the flights in order to limit the dispersion of points related
to atmospheric heterogeneity on leveled ULA legs. As observed in earlier studies, the dataset is clustered along a typical mixing
line in $\delta^2H - q$ space (Noone, 2012; Salmon et al., 2019; Sodemann et al., 2017). The end-members of the mixing lines show
substantial day-to-day variations, from $\delta^2H$=-110 to -80‰ (~[-16, -12]‰ for $\delta^{18}O$, not shown) for the more humid end
member (q > 8 g kg⁻¹), and from $\delta^2H$=-340 to -230‰ (~[-30, -20]‰ for $\delta^{18}O$, not shown) for the drier end member (q < 3 g
kg⁻¹). It is important to keep in mind that both, the variation in maximum flight altitudes and the meteorological situation can
contribute to variability.





As shown in Fig. 16b, the vertical profiles of the isotope content $\delta\,^2H$ are mostly uniform below 2 km, and partly have strong vertical gradients at higher elevations. The same patterns are observed on the vertical profiles of $\delta^{18}O$ (not shown). Strong gradients are present during flights 5 to 10 (16 to 18 June), with $\delta\,^2H$ changing rapidly within a few hundred metres of elevation. Most other flights revealed rather well-mixed conditions, such as during flights 13 to 15 (19 to 20 June), with the highest elevation of ULA-IC of 4.7 km reached during flight 13 (Table B2 of Appendix A).

Vertical gradients in isotope ratios changed according to the weather evolution, among other factors by modifying the stratification of the lower troposphere above the valley. After intense thunderstorms in the morning of 16 June, the initial strong vertical gradient in $\delta\,^2H$ at 2.5 km, also observed during both 16 June afternoon (Flight 5) and 17 June morning (flight 6), reduced substantially during 17 June afternoon (Flights 8-9). Large vertical gradients are also observed at higher altitudes (between 3 and 3.5 km a.m.s.l.). Such a variability is probably specific to the valley being embedded in low mountain ranges.

The main vertical structures are derived from the vertical profiles of potential temperature, aerosols, wind and relative humidity (RH). For the illustration, Fig. 17 shows a typical vertical RH profile that highlights the different layers present between the lake (~0.5 km a.m.s.l.) and about 4 km a.m.s.l.: the lake boundary layer between ground level and 1 km a.m.s.l. (layer 1), an atmospheric layer between about 1 and 2.5 km a.m.s.l. influenced by the area of the lake surrounded by mountains about 2-2.5 km high on its southern part (layer 2), a transition towards the free troposphere between about 2.5 and 3.5 km, influenced by the regional circulation (layer 3) and above 3.5 km a.m.s.l. the free troposphere with a synoptic influence, where long-ranged transport of air masses may occur.

The gradient in isotope content was related to a strong inversion between layers 2 and 3 where some clouds can be observed (see Fig. 11a), and was at least partly related to the advection and descent of dry air masses from the Libyan anticyclone during the 13-14 June period (Fig. 8a). The near-complete vertical mixing, and thus the uniform isotope content near the surface are testimony to active turbulence above complex terrains during summer. In contrast, the strong gradients above 2.5 km a.m.s.l. are probably due to regional advection, including the descend of airmasses from higher altitudes, as observed in the airborne observations of the water vapour isotopes in the Mediterranean (Sodemann et al., 2017) during the Hydrological cycle in the Mediterranean experiment (HyMeX). Above 3 km a.m.s.l., a humidity gradient is also observed and the formation of different types of clouds at the interface of the layer 3 and the free troposphere has been observed many times during the field campaign (see Fig. 11a). These gradients are related to strong contrasts in air mass transport.

### 5.2.2    Cloud liquid water sampling

Four relevant cloud water samples have been taken with CASCC during 18, 20 and 21 June (Table B1). The isotope content of the cloud water samples are shown in Fig. 18a (coloured circles). They are close to the global meteoric water line (GMWL) with corresponding d-excess values ranging from 12.1 to 14.8 ‰. The cloud water sample taken in the morning of 20 June (Fig. 18a, orange symbols) carries a weak evaporation signal. Equilibrium condensates were calculated from water vapour isotope measurements during the time when the cloud samples were taken (Fig. 18a, coloured squares) using the fractionation





factors of Majoube (1971), and air temperature measurements at cloud level, ranging between -4 and 1°C. It is worth noting that given potential sources of uncertainty, the equilibrium condensate values agree remarkably well with the cloud water samples during 18 June (blue symbols), and the afternoon of 20 June (green symbols). Overall, results here confirm that the

cloud water formed from equilibrium fractionation from ambient vapour. It is worth mentioning that such agreement between the completely independent sampling and measurement of vapour and cloud water supports the overall validity of i) the cloud water sampling protocol, ii) the airborne vapour isotope measurements and iii) the consistency of the calibrated water isotope dataset derived from the L-WAIVE campaign.

### 5.2.3    Precipitation sampling

In total, 28 precipitation samples, of which 22 are unique, have been taken during the campaign (see Table B1), with sampling times lasting from 20 min to several hours, depending on rainfall rate. The isotope composition in rainfall ranges from ~-11.2 to 2.2‰ in $\delta^{18}O$ and -73.5 to 9.6 ‰ in $\delta^{2}H$ (Fig. 18a, grey triangles), with corresponding values of d-excess between ~3.5 and 20.8‰ (not shown). Replication of measured isotope values on duplicate samples confirm complete sample preservation. Notably, there is an overall correspondence between the isotope range observed in cloud water samples and in a majority of

precipitation samples. Deviations of precipitation samples from the GMWL indicate potential below-cloud exchange, or post-condensational exchange processes leading to an enrichment of the water drop in $^{18}O$ (Graf et al., 2019; Worden et al., 2007). Samples taken between 12 to 15 June, and partly on 21 June are most enriched in $^{18}O$, exhibiting higher values in $\delta^{18}O$ and $\delta^{2}H$ and a calculated smaller d-excess (even negative ~-8‰). These samples are from rainfall events associated with local thunderstorms (see Fig. 11a). The low d-excess of these samples may point out the evaporation of rain droplets during their

fall. The less negative isotope values indicate exchange between falling raindrops and ambient water vapor below cloud base. Even if evaporation effects from the sampling setup cannot be fully excluded, in particular for the samples with a sampling duration of more than 3h, relatively high d-excess values during most samples, and the consistency of duplicate samples indicate that the influence of sampling artefacts can overall be considered as secondary.

### 5.2.4    Lake liquid water sampling

Evaporation from the Annecy lake is expected to be an important source for the water vapour in the Annecy valley. In order to link the atmospheric profiles of water vapour isotopes to the lake as a moisture source, 20 lake water samples (Table B1) were taken throughout the campaign within the lake-atmosphere interface layer (6 samples), as well as between 0.1 and 2 m depth (14 samples), and analysed for their stable water isotope composition. The average isotope contents were -8.3±1.5‰ for $\delta^{18}O$, and -63.0±6.0‰ for $\delta^{2}H$. Lake water samples taken close to the surface are expected to be most affected by evaporation,

causing deviations to the right from the GMWL along evaporation lines (enrichment in $^{18}O$). This is clearly observed in Fig. 18b for the majority of the samples. The corresponding d-excess underline such a result for the 6 lake water samples from the lake-atmosphere interface layer, with values ranging between 2.2 and -19.6‰ and an average value of -6.3‰. For the





samples taken at a depth of 10-20 cm on one day (Fig. 18b, square symbols), the derived d-excess showed less evaporation influence, with a median of 6.3‰, and a standard deviation of 6.3‰. The fact that there are differences between the isotope

contents at the interface and between the uppermost layer and 2 m depth points to incomplete mixing due to surface winds. Indeed, temperature profiles taken within the lake show a typically strong, but variable thermocline at about 5-8 m depth during the campaign (Fig. 7). While some sampling artifacts cannot be fully excluded, the influence of evaporation appears to decrease towards a depth of 2 m.

To assess the coherence between the lake water and the water vapour isotope composition measured by ULA and boat, we

calculated the isotope composition of liquid condensate resulting from an equilibrium fractionation process (Fig. 18b, triangles). The range of equilibrium condensate $\delta\,^2H$ values from the ULA (triangles) matches overall with the range of $\delta\,^2H$ observed in lake water for 16, 17 and 21 June (dots, squares). The $\delta^{18}O$ in equilibrium condensate is substantially more depleted, confirming the existence of kinetic (non-equilibrium) fractionation during lake evaporation. During 18 and 22 June, equilibrium condensate is more enriched than lake water, pointing to the influence of other sources that contribute to the water

vapour isotope composition. It is worth noting that Lake Annecy is fed by a catchment whose surface is 10 times larger than that of the lake via ten main tributaries located on the lake's periphery. The flows of its tributaries increase significantly in situations of heavy rain, which can substantially influence the isotopic content of the lake during the course of a year. This first-order comparison provides the basis for a more thorough analysis on how the lake evaporation signal is imprinted on the atmosphere above the lake in forthcoming studies.

**5.2.5    Synthesis and discussion**

For the sampling environments illustrated in Fig. 17, the overall results for stable isotopes in water are summarized in Fig. 19 on a statistical basis. This synthesis is presented in the form of whisker boxes for d-excess (Fig. 19a), $\delta\,^2H$ (Fig. 19b) and $\delta^{18}O$ (Fig. 19c). For each sub-figure, the upper part depicted the statistical content of isotope in water vapor and the bottom part in the liquid water for each sampled reservoir. For each whisker box, the central continuous vertical line indicates the

median (50th percentile, $P_{50}$), and the bottom and top edges of the box indicate the 25th ($P_{25}$) and 75th ($P_{75}$) percentiles, respectively. The whiskers extend to the most extreme data points not considered outliers, which are values outside the interval bounded by the 5th and 95th percentiles.

For the atmospheric water vapor, the sample number $n$ is significant (numbers given at the right of the whisker boxes in Fig. 19b) and it makes sense to accurately assess the confidence interval at 95% defined by $\left[\left(P_{50} - 1.57 \cdot\right.\right.$

$\left. (P_{75} - P_{25})/\sqrt{n}\right)\ \left(P_{50} + 1.57 \cdot (P_{75} - P_{25})/\sqrt{n}\right)\right]$. The notches in the upper part of the sub-figure represent this confidence interval. As they do not overlap from an atmospheric layer to another, we can conclude, with 95% confidence, that the true medians do differ. Hence, we highlight a significant variability in the stable isotope content of water vapor depending on the atmospheric layers previously identified.



Between the surface of the lake and 2.5 km a.m.s.l., we find $\delta^{18}O$ values similar to those recorded by Craig and Gordon (1965) for evaporation over the Mediterranean [-15,-10] ‰. On the other hand, they report much more dispersed d-excess values ranging from 5 to 35‰. Gat (2000) gives narrower values for marine and European air masses with a d-excess interval of [7-11] ‰. Our observations are mainly outside this interval for the first two layers, they overlap it for the 2.5-3.5 km a.m.s.l. layer more influenced by both regional and long-range transports.

Our liquid water samples are not numerous enough to define a confidence interval. As explained previously, we nevertheless note an enrichment in heavy isotope of the surface layer of the lake which is directly in contact with the atmosphere and therefore directly subject to evaporation. The deep layers are significantly less enriched and their isotope content ($\delta^{18}O \sim$-9‰ and $\delta^{2}H \sim$-65‰) is close to that given by Jasechko et al. (2013) for samples from Lake Superior (USA) which is not a high-altitude lake but is close to mean sea level. In addition, measurements of $\delta^{18}O$ profiles in the Annecy lake have already been carried out between the years 2000 and 2002. They have shown $\delta^{18}O$ values of the order of -9.5‰ for water above the
thermocline of "Petit Lac" during the summer months (Danis, 2009), in agreement with our results. This indicates a year-to-year stability of the isotopic content of the lake water below the surface microlayer.

To our knowledge, this is the first time that in-situ samples of cloud water and water vapour measurements have been taken directly in clouds. From a first analysis, cloud water formed in an equilibrium fractionation process for the samples collected here, and the cloud water isotopes were relatively similar to local precipitation (taken on other days). Since 1992, the Global
Network of Isotopes in Precipitation (GNIP, https://nucleus.iaea.org/wiser) regularly sample rainwater at the Thonon-les-Bains city (60 km northeast from Annecy) for isotope analyses. Based on data available until 2018, typical summer $\delta^{18}O$ ($\delta^{2}H$) is within the interval [-16, -1]‰ ([-120, -20]‰ ), encompassing our precipitation and cloud water measurements.





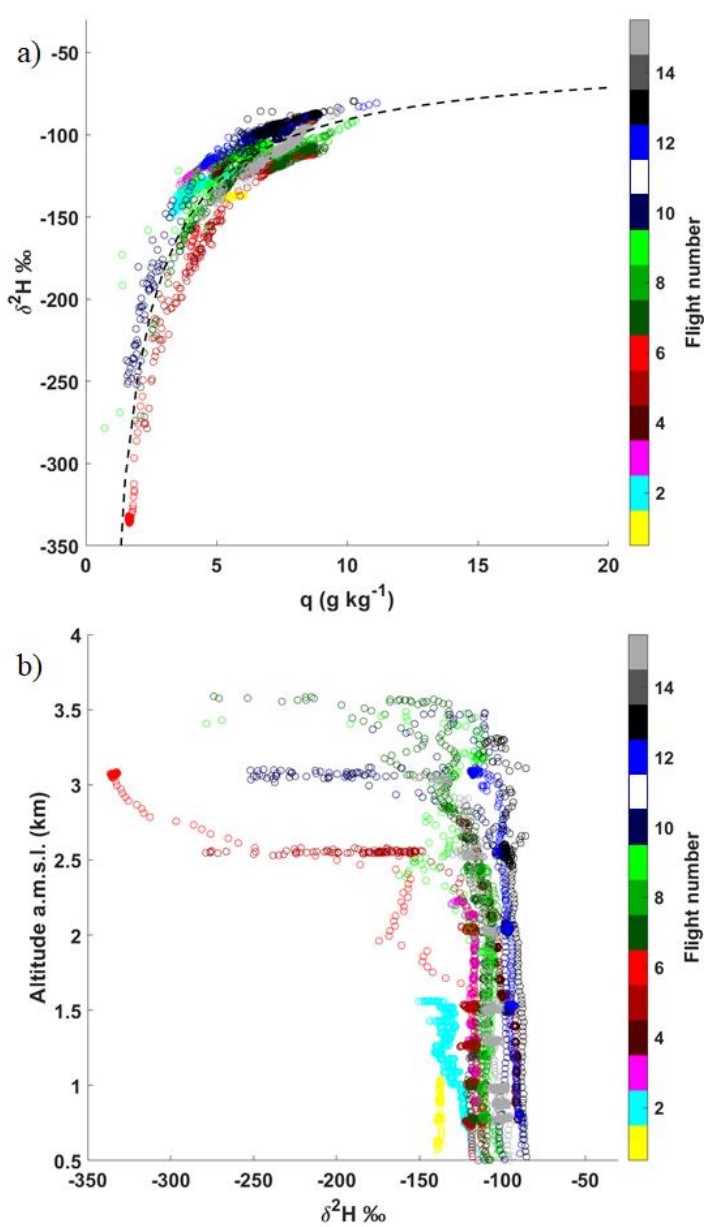

**Figure 16. Overview of airborne water vapour isotope measurements. (a) Isotope content $\delta^2H$ (permil) vs. specific humidity (g kg⁻¹). The mean mixing lines is reported using a black dotted line. (b) Vertical profiles of $\delta^2H$ (‰). Color indicates flight number. Isotope data are averaged at 10s interval. Only data points during ascent are displayed.**






**Figure 17. Schematic representation of the vertical structuring of the lower troposphere above the "Petit lac d'Annecy". The reservoirs where gaseous and liquid water samples have been taken are highlighted. A typical vertical profile of atmospheric relative humidity is reported.**





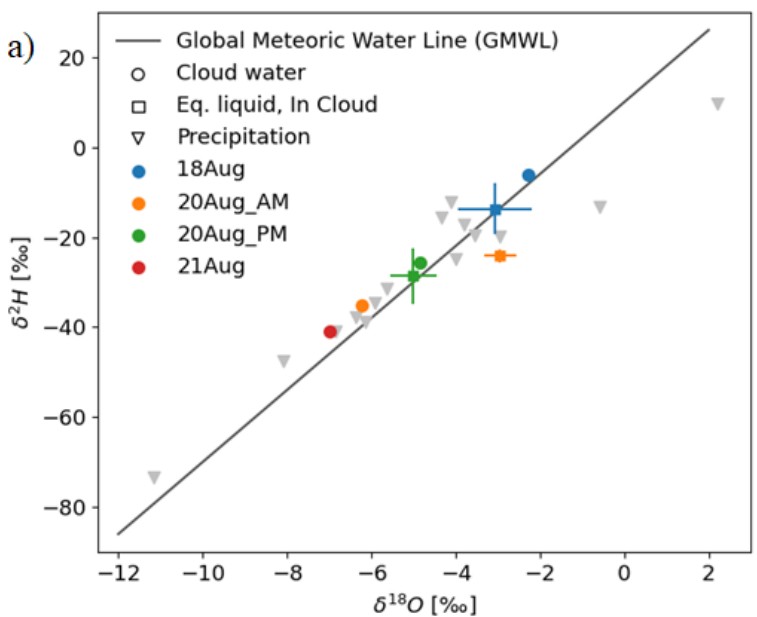

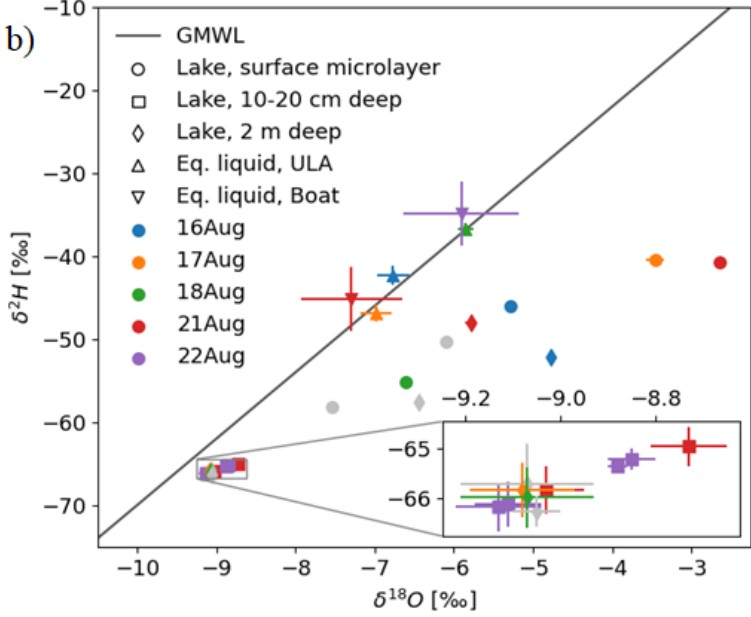

**Figure 18. Comparison of liquid samples with equilibrium condensate from vapor isotope measurements. (a) Cloud water samples (circles), equilibrium condensate (squares), and precipitation samples (grey triangles) a $\delta^2H$ - $\delta^{18}O$ plot. (b) Lake water samples at different depth compared to equilibrium condensate from vapour isotope measurements from ULA (upward triangle) and boat (downward triangle). Color denotes matching dates. Black line denotes the Global Meteoric Water Line (GMWL).**





**Figure 19. Whisker boxes for a) d-excess, b) $\delta^2H$ and c) $\delta^{18}O$. For each sub-figure, the upper part depicted the statistical content of isotope in water vapor and the bottom part in the liquid water.**





## 6    Conclusion

An overview of the field campaign L-WAIVE has been presented. Within L-WAIVE, we have developed an innovative approach to study evaporation processes and their heterogeneity over the Annecy lake. The field campaign took place between 12 and 23 June 2019 and used the complementary and/or synergistic measurements from different means of sampling gaseous 625 and liquid water. Continuous high vertical resolution profiling of tropospheric water vapour, temperature, and wind as well as aerosols in the Annecy valley have been performed. They are recorded together with ship-borne and airborne measurements of stable water isotopes ($H_2O$, $H^2HO$ and $H_2^{18}O$) in the lake, in the lower atmosphere as well as in precipitation and and in-cloud using an original sampling method from ULA.

The spatiotemporal evolution of the vertical profiles of WVMR appear coherent between the lidar retrievals and the ERA5 630 reanalysis, as well as with the measurements of the meteorological probes on board the ULAs. A marked diurnal cycle is observed with maximum WVMRs during the day (between 8 and 11 g kg$^{-1}$) when the valley winds are advected above the lake before reaching the measurement site. The vertical profiles of isotope composition are shown to be mostly uniform below 2 km a.m.s.l. (~1.5 km a.g.l.), and to exhibit strong vertical gradients at higher elevations, with a marked decrease against altitude at the interface between the different layers identified above the valley. We note that the cloud water samples are close to the 635 GMWL, (d-excess between 12.1 and 14.8 ‰), which indicates that the cloud water formed from equilibrium fractionation of ambient atmospheric water vapour. There is an overall correspondence between the isotope range observed in cloud water samples and a majority of precipitation samples (d-excess between -8.6 and 20.8/‰). The deviations observed on precipitation samples that are below the GMWL indicate potential below-cloud exchange, or post-condensational exchange processes. The average isotope composition of the lake water, taken at 2 m depth appears different from the one for the surface microlayer 640 sample, due to evaporation processes. Moreover, the $\delta^{18}O$ in equilibrium condensate above the lake is generally substantially more depleted, confirming the existence of non-equilibrium fractionation during lake evaporation. It is worth noting that during 18 and 22 June, equilibrium condensate was calculated as more isotope-enriched than Petit Lac water, pointing to the influence of other sources on the water vapour isotope composition above the lake.

The notable atmospheric vertical gradients of stable isotope composition, and day-to-day variation throughout the 645 measurement campaign were clearly related to the current weather conditions, modified by local topography. Beyond the value of the dataset from L-WAIVE for satellite remote sensing validation, such as the Sentinel-5P satellite (https://sentinel.esa.int/web/sentinel/missions/sentinel-5p), similar experiments will be useful to test the dynamic range and resolution of $H^2HO$ by remote sensing instrumentation, including ground-based and space borne LIDAR.

**Author contributions.** Patrick Chazette coordinated the field campaign and wrote the paper. Cyrille Flamant, Harald 650 Sodemann, Anne Monod, Elsa Dieudonné and Julien Totems contributed to the paper writing. Harald Sodemann and Andrew Seidl calibrated and derived the isotope data. Julien Totems calibrated WALI and derived the water vapor profiles. Patrick Chazette, Harald Sodemann, Julien Totems, Anne Monod, Elsa Dieudonné, Alexandre Baron, Andrew Seidl, Pascal Doira,





Amandine Durand, and Sylvain Ravier participated to the field campaign. Hans Christian Steen-Larsen and all the authors have proofreading the paper.

**Competing interests.** The authors declare that they have no conflict of interest.

**Acknowledgments.** Friendly acknowledgements to local authorities of the town of Lathuile R. Aumaître, H. Bourne, F. Lambert; instrument providers F. Arthoud (USMB); D. Crucciani for its welcome at the Delta Evasion airfield; ULA pilots F., L., F. Toussaint; technical assistance F. Maignan and C. Diana. This research was funded by the WaVIL project (WAter Vapor and Isotope Lidar, Agence Nationale de la Recherche grant n°ANR-16-CE01-0009). HS and AS received funding from the
Research Council of Norway with the FARLAB project (Contract no. 245907) and the ERC-2017-CoG ISLAS (Grant Agreement Nr. 773245). The IAEA/WMO Global Network of Isotopes in Precipitation (GNIP) is acknowledged for the access to its database accessible at: https://nucleus.iaea.org/wiser.

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



## Appendix A: L-WAIVE data availability

**Table A1. Data available during the period of the field experiment between June 11 and 23, 2020. The "V" indicates the half days of flight. The cross "X" indicates that the instrument was operated on a half-day basis. The malfunctions are denoted by italics "*V*" and "*X*". The number of flights is indicated per ULM during each half-day. For in situ samplings, the number of samples per half-day is indicated. During the first two flights of the ULA-IC, the iMet sonde was not operating (denoted by italic number). A sequence of ULA-IC flights experienced saturation in the inlet, that requires data filtering (denoted by bold number).**

| Instrument | | 11 | 12 | 13 | 14 | 15 | 16 | 17 | 18 | 19 | 20 | 21 | 22 | 23 |
|---|---|---|---|---|---|---|---|---|---|---|---|---|---|---|
| **Ground-based measurements** | | | | | | | | | | | | | | |
| Lidar WALI | | | X X | X X | X X | X X | X X | X X | X X | X X | X X | *X* | | |
| Wind lidar WLS100 | | | | | X X | X X | X X | X X | X X | X X | X X | X X | X X | X X |
| Rainwater sampling | | 1 | 3 | 1 | 2 | 2 | 1 | 7 | | | 3 | 6 | 1 | 1 |
| **Airborne measurements** | | | | | | | | | | | | | | |
| ULA-A | | | V | V | V | V | | V | V | V | V | V | V | V |
| | ALiAS | | | 1 | 1 | 2 | | 1 | 1 | 2 | 2 | 2 | 1 | 1 |
| ULA-IC | | | V | V | V | V | | V | V | V | V | V | V | |
| | CRDS | | | *1* | *1* | 1 | | 1 | 1 | 1 | **2** | 1 | 1 | **2** |
| | CASCC | | | | X | | | X | | | X | X | X | X |
| UAV | | | 1 | 1 | | | 1 | | 1 | 1 | 1 | | 1 | |
| **Boat measurements** | | | | | | | | | | | | | | |
| Sounding | | | 1 | | | | 1 | 2 | 2 | 2 | 1 | 1 | 1 | 1 |
| Lake sampling | | | 2 | | | | 2 | 2 | 2 | 1 | 2 | 1 | 2 | |
| CRDS | | | | | | | | | | | | | X | X |






## Appendix B: Ultra-light aircraft flights description

**Table B1. Flights characteristics for the remote sensing payload (ULA-A).**

| Flight | Date & time (LT*) dd/mm HHMM-HHMM | Maximum flight altitude during slow spiral ascent (km a.m.s.l.) | Comment |
|---|---|---|---|
| F01 | 12/06 1845-1850 | - | Aborted |
| F02 | 13/06 1000-1200 | 3.5 | Profile |
| F03 | 13/06 1800-1930 | 3.8 | Profile & transverse |
| F04 | 14/06 1630-1700 | 2.5 | Profile |
| F05 | 14/06 1730-1830 | 4.1 | Profile |
| F06 | 16/06 1030-1100 | 3.5 | Profile |
| F07 | 16/06 1700-1800 | 3.5 | Profile & lake axis |
| F08 | 17/06 0900-0940 | 3.4 | Profile |
| F09 | 17/06 1030-1150 | 3.7 | Profile & transverse |
| F10 | 17/06 1545-1700 | 3.7 | Profile & transverse |
| F11 | 17/06 1745-1900 | 3.7 | Profile & transverse |
| F12 | 18/06 0830-0945 | 3.7 | Profile |
| F13 | 18/06 1100-1230 | 3.7 | Profile & transverse |
| F14 | 18/06 1550-1715 | 4.1 | Profile & transverse |
| F15 | 19/06 0845-1005 | 3.5 | Profile & transverse |
| F16 | 19/06 1115-1250 | 4.5 | Profile |
| F17 | 19/06 1535-1720 | 3.7 | Profile & other valley |
| F18 | 20/06 1210-1335 | 4.3 | Profile & lake axis |
| F19 | 20/06 1620-1725 | 4.0 | Profile & lake axis |
| F20 | 21/06 1130-1240 | 3.5 | Profile & lake axis |
| F21 | 22/06 1100-1200 | 4.1 | Profile |
| F22 | 22/06 1600-1650 | 3.6 | Profile |
| F23 | 23/06 1120-1330 | 4.2 | Profile & other valleys |

*Local Time (LT)

**Table B2. Flights characteristics for the in situ payload (ULA-IC).**

| Flight | Date & time (LT*) dd/mm HHMM-HHMM | Maximum flight altitude during slow spiral ascent (km a.m.s.l.) | Comment |
|---|---|---|---|
| F01 | 12/06 19:23-19:50 | 1.0 | Local survey Malfunction of the iMet |
| F02 | 13/06 10:25-12:50 | 1.6 | Lake survey Malfunction of the iMet |
| F03 | 13/06 18:22-19:49 | 2.2 | Lake survey |
| F04 | 14/06 16:53-18:49 | 2.8 | Lake survey |
| F05 | 16/06 16:30-18:44 | 2.6 | Lake survey |
| F06 | 17/06 10:13-11:20 | 3.1 | Lake survey |
| F07 | 17/06 16:35-18:20 | 3.6 | Profiles |
| F08 | 18/06 10:30-11:31 | 2.4 | Profiles |
| F09 | 18/06 11:56-12:51 | 3.5 | Lake survey |
| F10 | 18/06 16:41-18:10 | 3.8 | Clouds |



| | | | Potential saturation point |
|---|---|---|---|
| F11 | 19/06 10:34-11:45 | 2.9 | Profiles<br>Potential saturated points |
| F12 | 19/06 15:39-16:58 | 3.1 | Lake survey<br>Potential saturated points |
| F13 | 19/06 17:49-19:18 | 4.7 | Clouds<br>Potential saturated points |
| F14 | 20/06 11:04-12:39 | 3.5 | Profiles<br>Potential saturated points |
| F15 | 20/06 14:49-17:16 | 3.4 | Lake survey |

*Local Time (LT)





### Appendix C: Lidar products description

The volume depolarisation ratio (VDR) is calculated following Chazette et al. (2012) for both lidars, WALI and ALiAS. It is retrieved with an absolute uncertainty of 0.2%. The VDR allows to highlight the particles that depolarize the radiation of the laser beam. These are particles that are not spherical, such as dust-like aerosols. The higher the VDR, the more the shape of the scatterers (molecules or aerosols) differs from a perfect sphere. The value of the molecular VDR corresponding to the WALI and ALiAS lidars is around 0.39% indicating that molecules cannot be considered as spherical scatters because their diffusion indicator is not perfectly isotropic. We use here the apparent scattering ratio ASR rather than the apparent backscatter coefficient because it is an indicator of scattering aerosol layers thanks to the molecular scattering correction. The ASR is given against the altitude $z$ by the relationship

$$ASR(z) = \left(1 + \frac{MBC(z)}{MBC(z) + ABC(z)}\right) \cdot e^{2 \cdot \overbrace{\int_{z_G}^{z} MSC(z') \cdot dz'}^{MOT(z)}} \qquad (C1)$$

where MBC and MSC are the molecular backscatter coefficient and molecular scattering coefficients, respectively. MOT is the range-resolved molecular optical thickness and ABC is the aerosol backscatter coefficient.





**Appendix D: Water vapor stable isotopes interpretive framework**

The isotope composition $\delta\ ^2H$ and $\delta^{18}O$ are given by the relationships

$$\begin{cases} \delta\ ^2H = {R\ _{^2H}}\big/{R^S_{^2H}} - 1 \\ \delta^{18}O = {R_{^{18}O}}\big/{R^S_{^{18}O}} - 1 \end{cases} \tag{D1}$$

Where $R^S_X$ is the isotope ratio of Vienna Standard Mean Ocean Water (VSMOW) for the compound X and $R_X$ the isotopic ratio given by

$$\begin{cases} R\ _{^2H} = {H\ ^2HO}\big/{H_2O} \\ R_{^{18}O} = {H_2\ ^{18}O}\big/{H_2O} \end{cases} \tag{D2}$$

The d-excess ($d$) is therefore defined as

$$d = \delta\ ^2H - 8{\cdot}\delta^{18}O \tag{D3}$$

to measure the deviation from equilibrium fractionation, noting the global average value for precipitation $d = 10‰$ (Dansgaard,
810     1964).