# Peer review of "The lacustrine-water vapor isotope inventory experiment L-WAIVE"

_Atmospheric Chemistry and Physics, 2020_

## Referee Comment (RC1) · Anonymous Referee #1 · 7 Jan 2021

Summary: The authors of this manuscript describe a set of measurements from a 2-week field campaign investigating atmospheric water vapor and environmental conditions above a high-altitude lake in the French Alps. They used a combination of many different sensors from ground-based lidar to observe atmospheric water vapor structure, cloud properties, and aerosols to ultra-light airborne platforms equipped with lidar, solar flux sensors, and water vapor isotope analyzer. Additional water isotope measurements were made from collections of cloud water and lake water at various depths, including the surface layer. It is a diverse dataset with potential to inform studies of lake evaporation and atmospheric mixing processes and perhaps provide validation for satellite measurements.

General comments: While the potential of this dataset is very promising, the current

manuscript does not deliver much interpretation. The authors do characterize vertical profiles of water vapor isotopes, but they do not comment on their interactions with clouds and aerosols. Nor do they discuss lake evaporation and atmospheric advection processes. It is clear that the data interpretation work is underway and planned for a future publication. This manuscript appears to be mainly the dataset description of many different types of observations during an intensive field campaign and presents observed ranges of isotopic values for vapor and liquid samples.

Having said that, this reviewer is not sure whether it meets the scope of ACP. "The journal scope is focused on studies with general implications for atmospheric science rather than investigations that are primarily of local or technical interest." Perhaps this would be more appropriately published as a dataset rather than a research paper? I will defer to the journal editor on this particular question.

If this limited interpretation is within the scope of ACP, here are some suggestions for making the value more apparent to the readers. 1. Why is the water vapor field in mountainous regions important to study? Societal importance? Local moisture recycling? 2. The authors could highlight their analysis linking large scale circulation from ERA to local wind patterns rather than declaring the aim of this manuscript to "gain understanding on the vertical structure of atmospheric water vapour above mountain lakes and to assess the respective influence of evaporation and advection processes" which they do not deliver in this particular manuscript. 3. "The influence of the lake evaporation was mainly detected in the first 500 m of the atmosphere." What data did the authors use to support this conclusion? 4. "A CRDS water vapour isotope analyser performed measurements during one day at the end of the experiment just above the lake surface in parallel with the lake water sampling." Was this data omitted?

Specific comments: Line 63 – See also this paper on tall tower measurements: Griffis, Timothy J, Jeffrey D Wood, John M Baker, Xuhui Lee, Ke Xiao, Zichong Chen, Lisa R Welp, et al. "Investigating the Source, Transport, and Isotope Composition of Water Vapor in the Planetary Boundary Layer." Atmospheric Chemistry and Physics 16, no.

8 (2016): 5139–57. https://doi.org/10.5194/acp-16-5139-2016-supplement.

Line 115 – Is the cloud water sampler discrete or passive (integrating the entire flight)? I think later it's mentioned that the collector opens while in the clouds only. Line 150 – the ULA's flew to 4 km amsl! That's amazing! Do pilots need O2? Line 181 – what is ALiAS? Line 190 – flux instead of flow? Line 205 – more details about the particle sizer inlet? Line 215 – were calibration functions designed specifically for this instrument? How do they change with cell pressure? Did cell pressure change with altitude? Line 228 – Appears an accompanying data report is panned with further details of the data processing and calibration procedure. Lines 246-250 are repetitive Line 245 – plastic other than HDPE? Specify material. Line 325 – surface microlayer sampling was interesting. Is this method previously published? Line 329 – was the cross section within the lake (liquid) or above the lake (vapor)? Line 348 – cite IAEA recommendations Line 434 -WL? Line 451 - The influence of the lake is thus seen mainly up to altitudes between ~1 and 2 km a.m.s.l. during the day and significantly lower the rest of the time (âĽš 0.5 km a.m.s.l.). How do the authors make that conclusion? Line 525 – Nice validation description, but Fig 18 shows large differences on Aug 20 and no vapor on Aug 21? Line 559 – Would it make more sense to calculate vapor in equilibrium with the lake water? To show kinetic evap influence? Line 591 – Actual values for Lake Superior is a strange comparison here given it's in a completely different location.

Fig 13 and 14 - Why does the lidar data availability with altitude change over time?

Fig 15 – not easy to compare the 2 ULA platforms. Different plot needed to compare. I'm not sure these figures show the profiles very well. The authors could consider plotting altitude vs WVMR for individual times with 2 ULAs and Lidar measurements as separate lines/colors.

Fig 18 – 2m deep samples are heavier than 10-20 cm deep? Why such tight values at 10-20 cm?

[Figure]

2020.

---

## Referee Comment (RC2) · Anonymous Referee #2 · 4 Feb 2021

The paper poses a series of interesting questions to begin: "What is the role of lakes [in] the...water cycle...? What is the relative contribution of evaporation from the lake to the atmosphere...downwind...?" but quickly pivots away from these questions and does not return to them in the conclusions. Instead, the paper describes in detail the measurement platforms involved in the L-WAIVE experiment, provides an overview of the synoptic conditions from reanalysis, and gives readers a first look at the various types of measurements made. There were clearly a lot of exciting new data collected as part of L-WAIVE. Perhaps it would be most appropriate if the paper stated upfront that this is a campaign overview paper and that much of the investigation of lake evaporation will take place in future work.

Below, I have organized my comments into major comments, related to the scientific

analysis, interpretation, and discussion, and minor comments, related to clarity, grammar, and presentation.

Major comments

Section 4.1 argues that a high pressure ridge brought warmer temperatures to the study site on the 14th of June, and a surface low over the British Isles brought colder temperatures on the 17th of June. However, it is not clear from the reanalysis maps that this was the case. Figure 9 shows equivalent potential temperature, which is also a measure of moisture content, not strictly temperature. Meanwhile, Figure 8 shows the study site more squarely within the center of a high pressure ridge on the 17th compared to the 14th. The British Isles appear to sit under a low on both days. In addition to these concerns, I wonder whether it wouldn't be simpler to show maps for each of the "golden" days (approximately one week), since it was such a short experiment, rather than seeing geopotential height for some days and RH for others. That way readers could see the progression from one synoptic state to the next clearly. Space could be saved by combining maps, for example, by adding geopotential height contours to temperature shading.

The paper thoroughly describes measurement and collection methods for nearly all observations except the precipitation samples, which are mentioned near line 412. How were these collected and analyzed? On which days were they collected?

Section 5.2.1 argues that strong vertical gradients in dD were observed on flights 5-10 and well-mixed conditions during flights 13-15; however, it is not clear Figure 16 supports this. First, Fig 16b shows vertically uniform isotope ratios for nearly all flights within the bottom 2 km of the atmosphere. (Flight 2 seems to be the exception.) This would indicate well-mixed conditions at these lower elevations. Most flights then show a near-step change at some height above 2km, but the gradient is still hard to gauge because, in most cases, there is almost no data from the free troposphere. Thus, the isotopic change in the vertical cannot be quantified. Perhaps the clustering of points on

Fig 16a could be used to estimate the degree of mixing within the lower atmosphere, but the number of observations at each altitude would have to be taken into account. An important related question: if the atmosphere is essentially well-mixed up to about 2km, wouldn't lake evaporation also influence the atmosphere to that height? What evidence is there that evaporated moisture only reaches 500 m?

In further reference to "vertical gradients," is Section 5.2.1 essentially talking about the height of the well-mixed boundary layer changing with synoptic conditions? The Section suggests these "gradients" change with air mass transport, but isn't atmospheric stability the more fundamental factor?

In Section 5.2.4, it is not clear how equilibrium fractionation is being calculated for lake water. Is the assumption here that water vapor condenses onto the lake as dew? This seems unlikely as a dominant mechanism. Clarification would be appreciated. Also, the relative depletion of lake water on Jun 18 and 22 is attributed to another (external) source. Yet, couldn't the depletion be caused by upwelling of water from lower depths from within the lake?

Section 5.2.5 argues that this is the first paper to present in-situ samples of cloud water and water vapor in cloud. This is certainly not the case. Presumably, the paper intends to say "isotopic" samples. Even so, previous studies have reported in-cloud water isotopic measurements. Examples include Lowenthal et al. 2011 (https://doi.org/10.1016/j.atmosenv.2010.09.061) and Lowenthal et al. 2016 (https://doi.org/10.1175/JTECH-D-15-0233.1). Also, Noone et al. gave presentations at the 2012 and 2017 Fall Meetings of the American Geophysical Union on isotope ratios collected in cloud water during two aircraft campaigns: the NSF ICE-T mission (2011) and the NASA ORACLES mission (2016-2018).

Minor Comments

Abbreviating "Alpine mountain lakes," which is redundant in and of itself, seems unnecessary. "Alpine lakes" could easily be shortened to "lakes" without confusion. Similarly, I would not abbreviate "water vapor mixing ratios." There are enough other new acronyms associated with the various measurement platforms.

Line 66 - I don't understand this sentence or why tall towers would provide "incomplete information" about evaporation. Is the suggestion here that they are not tall enough?

Line 80 - No need for "so".

Line 85 - no need for "a" after "original."

Line 161 - "of" is required after "out."

Figure 2b - could the lines be colored by aircraft to distinguish the two flight patterns? What do the red and purple straight lines represent?

Line 206 - is "flow" the right word?

Line 268 - It appears there is some redundancy in the next few lines.

Line 379 - Values are given for the EVAP standard, why not for GSM1?

Line 423 - Substitute "gives" for "allows to get."

Line 440 - Should this read "at altitude?"

Line 441 - It's not clear how the labels "Rain" and "Thunderstorm" provide information about cloud type.

Line 446 - Should this read "above 3.5 km?"

Figure 11 - The white shading and labels make it difficult see the aerosol scattering ratio signals. Also, the caption says that dust and pollution aerosols are labeled, but where?

Line 469 - It would help to remind the reader of when the "dust transport" event occurred.

Figure 15 - Can the flight humidity profiles be overlain on top of the lidar (Figure 14a)

for easier cross-comparison?

Line 535 - What does "partly" mean here? Some of the time? Somewhat strong?

Line 635 - "reservoir" seems somewhat confusing here, as I'm not sure most readers would think of distinct atmospheric layers as being distinct reservoirs.

Line 642 - Could the confidence interval equation be referenced? Also, seeing where the notches end in Fig 19 is not easy.

Fig 16 - Is the "mean mixing line" actually a best fit through all observations?

Fig 17 - It is not clear to me where the highlighting is for the gas-phase and liquid water samples.

Fig 18 - perhaps add "estimates" after "equilibrium condensate"

Line 702 - Is it fair to say this is "local" water vapor, which could have a lake source?

Appendix A is not particularly intuitive. What do the numbers represent in the table?
* * *

---

## Editor Comment (EC1) · Heini Wernli (Editor) · 14 Feb 2021

Dear Patrick, dear co-authors

As you have seen, both reviewers have submitted their comments to your interesting isotope field experiment paper. Both reviewers emphasise the great value of this diverse dataset, however they also struggled a bit with understanding the objective of this first publication on the L-WAIVE field experiment. Reviewer 1 mentions the option to publish this study as a dataset paper. You decided against this and submitted your manuscript as a normal ACP paper. I think this is fine, but it requires you to provide more interpretation of the datasets in the revised version of your paper. Also reviewer 2 was expecting you to discuss the questions you pose at the beginning of your paper

in the light of the new observations. My recommendation therefore is that you perform major revisions of your manuscript, considering all the reviewers' comments and with a special focus on better framing the paper as (i) a campaign overview paper but (ii) also a paper that provides new insight into the many questions posed in the introduction, based on the interpretation of the new measurements.

I am looking forward to the revised version of your paper! With best regards, Heini

---

## Author Comment (AC1) · 29 Mar 2021

General comments: While the potential of this dataset is very promising, the current manuscript does not deliver much interpretation. The authors do characterize vertical profiles of water vapor isotopes, but they do not comment on their interactions with clouds and aerosols. Nor do they discuss lake evaporation and atmospheric advection processes. It is clear that the data interpretation work is underway and planned for a future publication. This manuscript appears to be mainly the dataset description of many different types of observations during an intensive field campaign and presents observed ranges of isotopic values for vapor and liquid samples. Having said that, this reviewer is not sure whether it meets the scope of ACP. "The journal scope is focused on studies with general implications for atmospheric science rather than investigations that are primarily of local or technical interest." Perhaps this would be more appropriately published as a dataset rather than a research paper? I will defer to the journal editor on this particular question. If this limited interpretation is within the scope of ACP, here are some suggestions for making the value more apparent to the readers.

**The paper is indeed an overview article, describing the overall objectives of the L-WAIVE campaign and detailing what we believe is an innovative and comprehensive experimental sampling strategy as well as the instrumental assets involved. There are plenty such papers in ACP. In addition, the paper also provides an overview of the measurements acquired during the campaign as well as provides the local and regional context in which they have been gathered. Preliminary highlights of some of the outstanding observations made during the campaign are presented as well as some interpretation of the results. Finally, the paper is meant to be a reference article on which future articles pertaining to case studies can build on (without having to describe at length the experiment and the instrumental assests), to inform broadly the scientific community and make it aware of the availability of the data existence.**

1. Why is the water vapor field in mountainous regions important to study? Societal importance? Local moisture recycling?

**The reason why it is important to study the water vapour field and the origin of its variability appears partly in the beginning of the introduction but can indeed be made more explicit. We have added a rationale for this at the beginning of the introduction:**
**"The regions surrounding mountain lakes are poorly documented, yet they are highly anthropized, as in the case of Alpine mountain lakes, which are suffering from significant ecological upheavals probably linked to climate change, which influences the rates of precipitation and ice melt. Water resources and biodiversity are under threat. Lake/atmosphere interactions significantly regulate the climatic conditions in the valleys. There is therefore an obvious societal importance in studying the undeniable role of this ecosystem on the water cycle, which makes a major contribution to maintaining biodiversity in these regions.**
**Why is the vertical structure of the water vapor field in the lower troposphere is only sparsely documented above Alpine lakes? …"**

2. The authors could highlight their analysis linking large scale circulation from ERA to local wind patterns rather than declaring the aim of this manuscript to "gain understanding on the vertical structure of atmospheric water vapour above mountain

lakes and to assess the respective influence of evaporation and advection processes" which they do not deliver in this particular manuscript.

**Indeed, we do not give, in this article, quantitative results on the origin of the water or on the influence of the evaporation of the lake. This work is in progress using appropriate modelling tools. We appreciate the reviewer's suggestion and we have modified the text in the introduction accordingly, to better highlight the link made in this article between large-scale and local circulation. This aspect was already mentioned in the introduction but perhaps not clear enough. We have also placed more emphasis on the originality of the approach using airborne observation. Finally, we have also highlighted the potential effect of the lake evaporation using examples in the end of Subsection 5.2.5 where the link with the local dynamic is now presented. Material summarizing these aspects has been added at the end of the abstract, in the form:**

**" A significant variability of the isotopic composition was observed within the first 1.5 km above ground level (a layer defined as the lake region of influence, below the average height of the mountains surrounding the lake) depending on weather conditions, as well as local and synoptic atmospheric circulations. We highlight that fairly well-mixed conditions prevailed in the lower free troposphere, between the lake surface and the first 1.5 km of atmosphere above it, except when the wind is very weak above the lake. In this case, a marked depletion in heavy isotopes is observed in the lake boundary layer. It is also shown that strong gradients of isotopic composition can be observed at higher altitudes depending on the mean mountains height, the vertical local stability and the synoptic circulation."**

3. "The influence of the lake evaporation was mainly detected in the first 500 m of the atmosphere." What data did the authors use to support this conclusion?

**The new discussion added in section 5.2.5. clarifies this aspect.**

4. "A CRDS water vapour isotope analyser performed measurements during one day at the end of the experiment just above the lake surface in parallel with the lake water sampling." Was this data omitted?

**These data have been added in Fig. 17 with the whisker boxes located at ~1m above the lake surface and presented in the associated text in Subsection 5.2.5.**

Specific comments:

Line 63 – See also this paper on tall tower measurements: Griffis, Timothy J, Jeffrey D Wood, John M Baker, Xuhui Lee, Ke Xiao, Zichong Chen, Lisa R Welp, et al. "Investigating the Source, Transport, and Isotope Composition of Water Vapor in the Planetary Boundary Layer." Atmospheric Chemistry and Physics 16, no. 8 (2016): 5139–57. https://doi.org/10.5194/acp-16-5139-2016-supplement.

**Thanks, the reference has been added (line 79 of the revised MS).**

Line 115 – Is the cloud water sampler discrete or passive (integrating the entire flight)? I think later it's mentioned that the collector opens while in the clouds only.

**Yes, the reviewer is right, the collector opens only in clouds. In Subsection 2.1, we have replaced " ...and to collect cloud water samples." by : "... and to collect cloud water samples. The latter were collected during specific cloud flights (when meteorologic conditions were favorable) and the cloud collector was opened only in clouds."**

Line 150 – the ULA's flew to 4 km amsl! That's amazing! Do pilots need O2?

**Yes, above 4 km a.m.s.l. the pilot uses $O_2$.**

Line 181 – what is ALiAS?

**The acronym is now defined at line 192 of the revised MS.**

Line 190 – flux instead of flow?

**Agreed, the correction has been done.**

Line 205 – more details about the particle sizer inlet?

**The inlet has been designed to ensure that at the relative air speed the sampling is as close as possible to isokinetic. The sampling head is quite classical, with a divergent, followed by a convergent. The inlet diameter is adapted to the flow rate of the device. This is a technical aspect which is out of the scoop of this article. A schematic view of the sampling probe is given hereafter:**

[Figure]

**As this is a classical design, we have not added this information in the paper. We are only proving the information for the sake of clarity with respect to the reviewer's comment.**

Line 215 – were calibration functions designed specifically for this instrument? How do they change with cell pressure? Did cell pressure change with altitude?

**Mixing ratio-isotope ratio dependency correction functions were specifically determined for this analyzer (ser no. HIDS2254), and checked repeatedly over time, as described in Weng et al. (2020). Under laboratory conditions, the measurement cell pressure is regulated to 50 +/- 0.02 Torr. We have included a figure in the appendix D that illustrates that measurements during the slow**

ascent profiles and level segments had increased variability, but no substantial, systematic changes in the cavity pressure.

Line 228 – Appears an accompanying data report is panned with further details of the data processing and calibration procedure.

**We have included Appendix D that provides details of the calibration procedure and instrument stability extracted from the above-mentioned data report.**

Lines 246-250 are repetitive

**The sentences have been rearranged as:**
**"A pre-cleaned Caltech Active Strand Cloud Water Collector (CASCC, Demoz et al., 1996) was mounted on ULA-IC, modified to efficiently collect cloud water (Fig. 3b) at the relative cruising speed of the ULA (85 to 100 km h-1). The CASCC was modified to efficiently collect cloud liquid droplets from the ULA."**

Line 245 – plastic other than HDPE? Specify material.

**We have replaced the sentence "All sampling materials used were plastic or HDPE" by: " All parts necessary to adapt the HDPE cones on the CASCC were made of plastic ".**

Line 325 – surface microlayer sampling was interesting. Is this method previously published?

**Yes, the method was published previously, and the reference (Cunliffe et al., 2013) is given in the text.**

Line 329 – was the cross section within the lake (liquid) or above the lake (vapor)?

**We have modified the sentence as " …the CRDS isotope analyser was taken on board the boat to sample a cross-section in the atmospheric layer just above the "Petit Lac…".**

Line 348 – cite IAEA recommendations

**The citation has been included as given below:**
**IAEA (2009): Reference Sheet for VSMOW2 and SLAP2 international measurement standards. Issued 2009-02-13, International Atomic Energy Agency, Vienna , 5 p., http://curem.iaea.org/catalogue/SI/pdf/VSMOW2_SLAP2 .pdf.**

Line 434 -WL?

**WL has been replaced by "wind lidar".**

Line 451 - The influence of the lake is thus seen mainly up to altitudes between 1 and 2 km a.m.s.l. during the day and significantly lower the rest of the time (âL'š 0.5 km a.m.s.l.). How do the authors make that conclusion?

**We have modified this part, which is now discussed more clearly in subsection 5.2.5. The effect of the lake is more easily observable in low wind conditions. We have made a better link with local dynamics by using lidar measurements.**

Line 525 – Nice validation description, but Fig 18 shows large differences on Aug 20 and no vapor on Aug 21?

**First, the 'Aug' tags were erroneous and have been modified to 'Jun' in the revised version of this Figure (now Fig. 15 in the revised MS). The morning flight**

**on 20 June was affected by a saturated inlet and should not be interpreted, the data point has been removed in the revised manuscript. The flight on 21 June took place without the vapour analyzer. This is now stated in the revised manuscript.**

Line 559 – Would it make more sense to calculate vapor in equilibrium with the lake water? To show kinetic evap influence?

**Yes, we have modified Fig. 18 (now Fig. 15 in the revised MS) and the comparison to show equilibrium vapour rather than condensate. We also have updated the corresponding text.**

Line 591 – Actual values for Lake Superior is a strange comparison here given it's in a completely different location.

**Yes, we have removed this comparison.**

Fig 13 and 14 - Why does the lidar data availability with altitude change over time?

**The range of lidars depends on atmospheric transmission and the presence or absence of clouds. If the cloud is too dense (i.e. has an optical depth in excess of ~3), then the laser beam is total extincted shortly after having penetrated the cloud and the atmosphere above the cloud base cannot be probed.**

Fig 15 – not easy to compare the 2 ULA platforms. Different plot needed to compare. I'm not sure these figures show the profiles very well. The authors could consider plotting altitude vs WVMR for individual times with 2 ULAs and Lidar measurements as separate lines/colors.

**It is true that this colour level figure does not sufficiently show contrasts. Moreover, we have already shown the correct agreement in Fig. 4 for a coordinated flight. The 2 ULMS did not always fly together and did not necessarily follow the same trajectories. This figure (Fig. 15 of the original MS) of limited interest has therefore been removed in the revised MS.**

Fig 18 – 2m deep samples are heavier than 10-20 cm deep? Why such tight values at 10-20 cm?

**The thermocline was measured at a depth of at least 7 m (Fig. 7 or the original MS, now Fig. 6). As we are interested in exchanges across the interface between the lake and the atmosphere, we wanted to check whether there were any notable differences between the surface waters and the waters located a little deeper, at a depth of around 2 m. The samples at 10-20 cm depth were only acquired during one day at the end of the campaign, unlike other liquid water samples. For the sake of not introducing a bias in the interpretation of the data we have decided to remove the time-limited dataset from Figure 17 (Fig.19 in the original MS).**

---

## Author Comment (AC2) · 29 Mar 2021

The paper poses a series of interesting questions to begin: "What is the role of lakes [in] the...water cycle...? What is the relative contribution of evaporation from the lake to the atmosphere...downwind...?" but quickly pivots away from these questions and does not return to them in the conclusions. Instead, the paper describes in detail the measurement platforms involved in the L-WAIVE experiment, provides an overview of the synoptic conditions from reanalysis, and gives readers a first look at the various types of measurements made. There were clearly a lot of exciting new data collected as part of L-WAIVE. Perhaps it would be most appropriate if the paper stated upfront that this is a campaign overview paper and that much of the investigation of lake evaporation will take place in future work.

**In the introduction, we mention the objectives of L-WAIVE and the questions/reflexions from which they stemmed in broader context. These must be differentiated from the objectives of the present article itself. Our intend here is to set the scene of forthcoming articles that will be dedicated to tackle in more details the questions posed in the Introduction. The purpose of the article is to present the experimental strategy designed for the campaign as well as the instrumental assets (and their complementarity) and to provide the first scientific highlights. We have modified the introduction to focus more on the objective of the article and to insist more on the links between the dynamics and the isotopic observations.**

**Major comments**

Section 4.1 argues that a high pressure ridge brought warmer temperatures to the study site on the 14th of June, and a surface low over the British Isles brought colder temperatures on the 17th of June. However, it is not clear from the reanalysis maps that this was the case. Figure 9 shows equivalent potential temperature, which is also a measure of moisture content, not strictly temperature. Meanwhile, Figure 8 shows the study site more squarely within the center of a high pressure ridge on the 17thcompared to the 14th. The British Isles appear to sit under a low on both days. In addition to these concerns, I wonder whether it wouldn't be simpler to show maps for each of the "golden" days (approximately one week), since it was such a short experiment, rather than seeing geopotential height for some days and RH for others. That way readers could see the progression from one synoptic state to the next clearly. Space could be saved by combining maps, for example, by adding geopotential height contours to temperature shading.

**We agree that the transition is not clear on the Figures provided in the original version of the MS. As proposed by the reviewer, we constructed a 12-day sequence (shown in Figs 8-10) with temperature, geopotential, and wind, all at 700 hPa which corresponds to the free troposphere just above the mountains. The text has been rewritten accordingly in section 4.1.**
**However, the potential temperature does not depend on water vapor, it is the equivalent potential temperature that depends on it.**

The paper thoroughly describes measurement and collection methods for nearly all observations except the precipitation samples, which are mentioned near line 412. How were these collected and analyzed? On which days were they collected?

**The reviewer is right, the collection methods for precipitation samples were omitted in the original version of the MS.**
**We have added a new section (Section 3.4) dealing with precipitation samples, in which we have added the following text:**
**"Precipitation samples were taken throughout the campaign period. The sampling device consisted in a pre-cleaned HDPE funnel directly connected to a pre-cleaned HDPE sampling bottle. The precipitation samples were manually operated: after each precipitation event, the sample was aliquoted into 1.5 ml glass vials with rubber/PTFE septa and stored at 4°C prior to 345 isotopic analysis, while the rest of the sample was stored at -18°C. Precipitation sampling times lasted from 20 min to several hours, depending on rainfall rate.".**
**The days on which they were collected are indicated in Table A1, where "Rainwater sampling" have been replaced by "Precipitation sampling".**

Section 5.2.1 argues that strong vertical gradients in dD were observed on flights 5-10 and well-mixed conditions during flights 13-15; however, it is not clear Figure 16 supports this. First, Fig 16b shows vertically uniform isotope ratios for nearly all flights within the bottom 2 km of the atmosphere. (Flight 2 seems to be the exception.) This would indicate well-mixed conditions at these lower elevations. Most flights then show a near-step change at some height above 2km, but the gradient is still hard to gauge because, in most cases, there is almost no data from the free troposphere. Thus, the isotopic change in the vertical cannot be quantified. Perhaps the clustering of points Fig 16a could be used to estimate the degree of mixing within the lower atmosphere, but the number of observations at each altitude would have to be taken into account. An important related question: if the atmosphere is essentially well-mixed up to about 2km, wouldn't lake evaporation also influence the atmosphere to that height? What evidence is there that evaporated moisture only reaches 500 m?

**Fig. 14b (in the revised MS, Fig. 16b in the original version of the MS) does not make it easy to follow the evolution of isotope abundance with altitude. We have moved the color contrasts. The figure has therefore been redrawn. The gradients in altitude are related to the transition to the free troposphere, and they are corroborated by the lidar observations (co-located aerosol backscatter gradients), but above all by the independent meteorological measurements on the two ULAs. The variation in altitude of the gradient is linked to the time of day when the flights were made.**

**We have modified the text:**

**"These strong gradients in altitude are related to the transition to the free troposphere, and are confirmed by the lidar observations and meteorological measurements on the two ULAs. The observed vertical gradients in isotope ratios evolved as a function of the different isotope composition of boundary layer vapor, free-troposphere vapor, and the stratification in the lower troposphere above the valley which depends on the meteorology (as for example rainy events) but also on the time of day when the flights were made. In the early afternoon, the boundary layer is shallower and less buoyant, leading to a transition at lower altitude (flights 5 and 6). Flights 5 and 6 also took place in the aftermath of heavy thunderstorms occurring in the morning of 16 June that led to a cooling of the surface. In the late afternoon, thermal convection is generally more efficient, and the transition may occur at higher altitudes (flight 10 on 18**

**June 1700 LT). Flight 9 on 18 June 1200 LT does not seem to follow the same rule. As shown in the vertical lidar profiles in Fig. 10, the clouds around 4 km a.m.s.l. seem to exert a forcing on the boundary layer which moves its top to higher altitudes (~3.5 km a.m.s.l.)."**

**What defines the different atmospheric layers over the valley are the vertical profiles of WVMR, potential temperature (even temperature), aerosol and wind. There is clearly a layer of at least 500 m above the lake that defines its influence on the atmosphere. Now, the reviewer is right that this influence is certainly not limited to this layer and evaporative water vapor can be moved higher in altitude depending on wind and vertical stability conditions. We have therefore reviewed this aspect in subsection 5.2.5.**

In further reference to "vertical gradients," is Section 5.2.1 essentially talking about the height of the well-mixed boundary layer changing with synoptic conditions? The Section suggests these "gradients" change with air mass transport, but isn't atmospheric stability the more fundamental factor?

**The explanation is given in the answer above. We agree with the reviewer that it is indeed the stability of the atmosphere that explains the location of these gradients at altitude. However, the advection of dry air at higher elevations probably enhanced gradients by transporting isotopically depleted air to the observation area on one of the days. This has been phrased more clearly in the revised manuscript.**

In Section 5.2.4, it is not clear how equilibrium fractionation is being calculated for lake water. Is the assumption here that water vapor condenses onto the lake as dew? This seems unlikely as a dominant mechanism. Clarification would be appreciated. Also, the relative depletion of lake water on Jun 18 and 22 is attributed to another (external) source. Yet, couldn't the depletion be caused by upwelling of water from lower depths from within the lake?

**Following the suggestion of Reviewer 1, we have changed this comparison to vapor from equilibrium evaporation from the lake on the one hand, and the vapor measurements from ULA and boat on the other hand. Changes in lake water composition can indeed play a role, and this has been included as a potential factor of influence in the revised manuscript.**
**Furthermore, the lake waters are highly stratified, and the temperature profiles show that there were no upwellings during the period of the experiment. The 'Petit Lac' is also little influenced by the tributaries and our measurements were taken in its center.**

Section 5.2.5 argues that this is the first paper to present in-situ samples of cloud water and water vapor in cloud. This is certainly not the case. Presumably, the paper intends to say "isotopic" samples. Even so, previous studies have reported in-cloud water isotopic measurements. Examples include Lowenthal et al. 2011 (https://doi.org/10.1016/j.atmosenv.2010.09.061) and Lowenthal et al. 2016 (https://doi.org/10.1175/JTECH-D-15-0233.1). Also, Noone et al. gave presentations at the 2012 and 2017 Fall Meetings of the American Geophysical Union on isotope ratios collected in cloud water during two aircraft campaigns: the NSF ICE-T mission (2011) and the NASA ORACLES mission (2016-2018).

**Indeed, the referee is right, we intended 'isotopic' samples. The 2 papers by Lowenthal et al. describe in-cloud isotopic measurements made in vivo using the instrumentation available at the permanent mountain-top facility of Storm Peak Laboratory in the Rocky Mountains at 3220 m above mean sea level. In this paper we describe isotopic samples that are made from an airborne facility. We were not aware of the AGU presentations given by Noone et al.**

**Minor Comments**

Abbreviating "Alpine mountain lakes," which is redundant in and of itself, seems unnecessary. "Alpine lakes" could easily be shortened to "lakes" without confusion. Similarly, I would not abbreviate "water vapor mixing ratios." There are enough other new acronyms associated with the various measurement platforms.

**Agreed, the correction was made for "Alpine lake" without introducing an acronym. On the other hand, for "water vapor mixing ratios", which is often used, we preferred to leave the acronym.**

Line  61-63 - I don't understand this sentence or why tall towers would provide "incomplete information" about evaporation. Is the suggestion here that they are not tall enough?

**The point is that tall towers provide point measurements at one or several heights, whereas airborne observations provide complete profiles over a height range. Moreover, there is no tower that reaches up to the free troposphere, except perhaps at the top of the mountain. On the other hand, it is usually not in the middle of the lake. It is therefore preferable to use a boat or a small aircraft such as an ULA.**

Line 80 - No need for "so".

**The correction has been done.**

Line 85 - no need for "a" after "original."

**The correction has been done.**

Line 161 - "of" is required after "out."

**The correction has been done.**

Figure 2b - could the lines be colored by aircraft to distinguish the two flight patterns? What do the red and purple straight lines represent?

**The two types of flights are already colored differently. An explanation has been added in the legend for the red straight lines and purple arrows.**

Line 206 - is "flow" the right word?

**The correction has been done.**

Line 268 - It appears there is some redundancy in the next few lines.

**The correction has been done.**

Line 379 - Values are given for the EVAP standard, why not for GSM1?

**The values for standard GSM1 have been added in the revised manuscript.**

Line 423 - Substitute "gives" for "allows to get."

**The correction has been done.**

Line 440 - Should this read "at altitude?"

**The correction has been done.**

Line 441 - It's not clear how the labels "Rain" and "Thunderstorm" provide information about cloud type.

**The sentence has been rewritten.**

Line 446 - Should this read "above 3.5 km?"

**No, larger particles are observed from ~2.5 km a.m.s.l. in Fig. 12.**

Figure 11 - The white shading and labels make it difficult see the aerosol scattering ratio signals. Also, the caption says that dust and pollution aerosols are labeled, but where?

**The white shading is used to show profiles influenced by clouds. The dusts and pollution aerosols are not labelled. They can be located using the VDR. The strong VDRs are associated with dusts and the weakest with pollution aerosols. The information has been added in the figure caption.**

Line 469 - It would help to remind the reader of when the "dust transport" event occurred.

**We have modified the sentences to link the dust presence to the date.**

Figure 15 - Can the flight humidity profiles be overlain on top of the lidar (Figure 14a) for easier cross-comparison?

**Fig. 15 has been removed as it does not provide significant information with respect to the lidar measurements and model outputs. The agreement between the measurements of the two ULAs is already shown in Fig. 4. It should also be noted that the ULA flights were not necessarily vertical to the lidar.**

Line 535 - What does "partly" mean here? Some of the time? Somewhat strong?

**The correction has been done. We have removed 'partly' from the sentence.**

Line 635 - "reservoir" seems somewhat confusing here, as I'm not sure most readers would think of distinct atmospheric layers as being distinct reservoirs.

**Yes, that is not an appropriate term and we have made the change: "…for each sampled altitude range in the atmosphere […] in the depth of the lake".**

Line 642 - Could the confidence interval equation be referenced? Also, seeing where the notches end in Fig 19 is not easy.

**The resolution of Fig. 18 has been degraded in the pdf, at high resolution you can easily see the limits of whiskers boxes. The definition of the confidence interval is quite classical and can be found in all statistics books. It is, among others, the definition adopted by the Matlab or Python environments. One is free to choose another criterion as long as it is clearly defined.**

Fig 16 - Is the "mean mixing line" actually a best fit through all observations?

**The mixing line has been computed using Noone (2012) and we have added the information in the caption of the figure.**

Fig 17 - It is not clear to me where the highlighting is for the gas-phase and liquid water samples.

**A clearer explanation was given in section 5.2.5 where the interest of measurements in the lake water and in the atmosphere is better highlighted.**

Fig 18 - perhaps add "estimates" after "equilibrium condensate"

**The caption of Fig. 18 (now 15) has been revised as**
**" Figure 1. $\delta\,^2H$ - $\delta^{18}O$ plots of liquid samples with the vapor isotope estimates. (a) Cloud water samples (circles), equilibrium condensate (squares), and precipitation samples (grey triangles). (b) Equilibrium vapor from lake water samples at different depth (dots and diamonds) compared to vapor isotope measurements from ULA-IC (upward triangle) and boat (downward triangle). Colour denotes matching dates. Gray colours show data where the vapor samplings are not available. Black line denotes the Global Meteoric Water Line (GMWL)."**

Line 702 - Is it fair to say this is "local" water vapor, which could have a lake source?

**It is not the water vapour that is said to be local, but the topography (modified by local topography).**

Appendix A is not particularly intuitive. What do the numbers represent in the table?

**Agreed. We have simplified the notations in the table and used colours to highlight special cases.**

---

## Author Comment (AC3) · 29 Mar 2021

**Editor comments**

As you have seen, both reviewers have submitted their comments to your interesting isotope field experiment paper. Both reviewers emphasise the great value of this diverse dataset, however they also struggled a bit with understanding the objective of this first publication on the L-WAIVE field experiment. Reviewer 1 mentions the option to publish this study as a dataset paper. You decided against this and submitted your manuscript as a normal ACP paper. I think this is fine, but it requires you to provide more interpretation of the datasets in the revised version of your paper. Also reviewer 2 was expecting you to discuss the questions you pose at the beginning of your paper in the light of the new observations. My recommendation therefore is that you perform major revisions of your manuscript, considering all the reviewers' comments and with a special focus on better framing the paper as (i) a campaign overview paper but (ii) also a paper that provides new insight into the many questions posed in the introduction, based on the interpretation of the new measurements.

**We have improved the article to make its purpose clearer and to add elements to better support what we have written about the influence of the lake on the lower tropospheric water vapor isotopic composition. We feel it is worth framing the 'overview' aspect of the paper because many components of the L-WAIVE experiment are quite original (use of multiple ULAs, ULA-borne CRDS measurements, isotopologues sampling in the vapor and liquid phase in the environment of the Annecy lake, …). Providing a comprehensive overview of the experiment is therefore important and requires that a significant part of the paper be dedicated to the underlying experimental strategy and the description of involved instrumental assets. This may have left the reviewers with the impression that the paper is merely a campaign report. We would like to insist that this is not the case and that the paper already contains a significant number of new insights, such as the consolidated vision of water isotopologues across the air/water compartments in a lake area. To further emphasize the 'new insights' aspect of the paper, a clearer link is made with the local, lower valley dynamics documented with the wind lidar measurements.**
**In the following, we provide a version of the article where all changes made to comply with the reviewer's comments are highlighted. Text in blue indicates added material, while text in green indicates text present in the original version of the MS that has been moved elsewhere in the revised version of the MS.**

---

## Author Response (AR2)

**Editor**

Many thanks for submitting a revised version of your paper about the fascinating L-WAIVE experiment. As you can see, the reviewer who asked for major revisions in the first round, mentions that the manuscript has improved, but fails in providing a consolidating vision, which is one of your main objectives. I also read your revised version and I agree with the reviewer that a further focusing and consolidation would greatly increase the value of this paper. My overall impression is that the paper tries to cover too many things and that leaving away certain aspects - those that are not essential for the water vapor isotope profiles - and discussing a bit more the key results could help a lot. Here a few more specific impressions I had when reading the paper:

1) Since I saw the paper for the first time, I found the title a bit cumbersome: to me, a simpler title like "A field experiment to investigate the vertical water vapor isotope profile above an Alpine lake" would be more appealing and informative.

**We changed the title to go in the direction of the editor. However, we also wanted to highlight that there are not only vertical profiles. Thank you for this suggestion which helps to better understand the scientific objectives of the article.**

2) I think your abstract has a rather weak ending. Lines 25-30, where you describe the first results from L-WAIVE contain very vague statements. In essence you write that there is a lot of variability depending on the synoptic circulation. To play the devil's advocate, you don't need to do a field experiment to come to this conclusion (this is obvious) ... your measurements and results contain much more precise and interesting information, and I would like to invite you to document some of them also in the abstract.

**The abstract has been revised to reflect the evolution of the article and to better highlight the scientific aspects that are discussed and the main results.**

3) I agree with the reviewer that the paper has a strange starting point (ice melt, biodiversity). These phenomena and concepts never appear again in the paper; I think you can directly start with humidity profiles etc.

**We fully agree and the 1st paragraph has been deleted.**

4) The title of the paper and the main objectives clearly focus on water vapor profiles and the stable isotope signals (which I think is perfectly fine!). However, the paper then also presents results about lake temperature profiles and aerosols, but I couldn't find out how these results influence your analysis of the vapor/isotope profiles. I rather felt distracted by the lake temperatures and aerosol results and had the impression that I was loosing track of the storyline. My suggestion is that you consider omitting these aspects of the paper and really focus on the instruments and data you need for the vapor profile analysis. This would also shorten the instrument description part.

**These elements were used to help interpret the water vapour data. The position of the thermocline is important to know the thickness of the layer where homogeneous mixing can be expected. The lidar atmospheric backscatter data are used to clearly position the rainfall periods in time, but also as a tracer of the vertical dynamics of the atmosphere.**

**We have removed Fig. 4 and used a reference to give the position of the thermocline of Lake Annecy in summer. Fig. 10 has been revised to make it clearer and to highlight the transition between the valley boundary layer and the free troposphere. This figure is important because it gives information on the temporal evolution of this transition zone which helps to explain the vertical water vapour profiles.**

**In addition, we have grouped all the profile-time evolution figures (Figs 12 and 13) in the same subsection because it is preferable to discuss them together in order to describe the observed atmospheric structures. The comparison to ERA5 has been removed as it did not add much to the discussion, which is mainly about local effects, the synoptic view having been given before, in subsection 4.1.**

**The text of the article has been amended accordingly and paragraphs have been moved.**

5) On a more detailed level, I think the paper could be shortened in several places. It sometimes reads a bit too much like a campaign report. For instance, in line 157 you mention "golden days", but then I could not find further analysis of these golden days. So why should the reader know about golden vs. silver days?

**We agree that these terms are not very appropriate, they have been removed. This is all the more important if it gives the impression that this article is a report. This paragraph has been removed as the information is already present in Appendix A.**

6) I also suggest to show less synoptic charts. They fill 3 pages but are not really discussed in great detail. Also, the link between a few selected charts and the analysis of the profiles could be strengthened.

**We agree with the editor and the reviewer. The multiple charts had been added at the request of one of the previous reviewers. Now, these 3 sets of figures have been reduced to one figure (Fig. 3) as there is not much variability in synoptic flows.**

7) I am not fully sure what I should think of the artwork in Fig. 16. On the one hand, I like schematics, but here too many things are unclear to me. For instance, what is the 0.46 km label refer to (altitude of the lake?). Is the lake boundary layer then extending to 1 km above the lake? The abstract emphasises an altitude of 1.5 km above ground level, is this 0.46+1 km? I am confused. And maybe more importantly, how can you show a typical profile of RH if conditions were so variable? And why is the cloud located where your RH profile has a minimum? I tend to suggest that you better omit this schematic or establish a (much) stronger link to the actual measurements.

**Fig. 16 was made to summarize the vertical structures encountered. It is true that they are quite fluctuating and that the figure is not representative of all situations. We have therefore removed this figure, which is not fundamental to the scientific objectives of the paper.**

Despite my critical remarks, let me conclude by emphasising that I congratulate the team for collecting an impressive and highly valuable data set (with a very interesting measurement strategy), and that I hope that the comments from the reviewer and myself help you to further improve the manuscript.

Your criticism is welcome and has helped us to refocus the article. Our vision was to give a broad view of L-WAIVE, but as you pointed out, this dispersed the objectives. This measurement campaign brought original data on different aspects and not all of them can be described in one article. Therefore, we have focused on the temporal and vertical evolutions in this article.

We have hence removed all the measurement tools that were not directly related to this new version or that were too general. Figures 3, 4, 5 and 11 have also been removed, along with the associated texts.

All this has led to the restructuring of some paragraphs, moving some parts, and completing others. Our objective has been refocused largely on the characterization of vertical profiles. We have therefore revisited paragraph 6 and added Fig. 8. This figure is important in this context because it shows that the vertical structures observed via isotopic measurements exist. It also allows a discussion on their evolution during the day and according to two contrasting local meteorological situations. We therefore thought that this would be a real plus for the scientific contribution of the article.

**Reviewer**

General comments:

The manuscript revision is improved and no longer over-sells in the intro and under-delivers in the interpretation. The editor's instructions were to:

"… perform major revisions of your manuscript, considering all the reviewers' comments and with a special focus on better framing the paper as (i) a campaign overview paper but (ii) also a paper that provides new insight into the many questions posed in the introduction, based on the interpretation of the new measurements."

The research objectives that the authors identify now are:
 "The main objective of this paper is to present a novel experimental approach to measure stable isotopes of water from the interior of steep valleys to the free troposphere in order to help identifying the origin of the air masses that contribute to the observed isotope ratios. " and "… proposes a consolidated vision of water isotopologues across the air/water compartments in a lake area."

The main results pertaining to these research objectives are in Fig 17 and there is a overview discussion of how the observations are different in the different layers, but this is far from providing new insight or a consolidated vision. Figure 16's atm structure diagram is very conceptual (at least the data that it's based on is not transparent). There is an attempt at some new interpretation, but this falls solidly in a campaign overview paper. Again, I leave the suitability of a campaign overview paper for ACP up to the editor.

**We searched the scientific literature for studies on the vertical distribution of water vapour in lake valleys. We did not find anything similar to the results of L-WAIVE. So there must be some originality in this work. We have refocused the paper and removed all elements that were not directly related to the vertical structure of the atmosphere and the evolution of stable water isotopes. We do not agree that this article is a campaign report. There is an initial description of the campaign and the observations because it is necessary to explain the basis on which the data were acquired before using them to interpret processes. The text has been reworked in some places and information moved to make the conclusions more prominent.**

**Fig. 16 has been removed as it is confusing, and we agree that it is not representative of the generality of encountered atmospheric situations.**

Specific comments:

The intro spin as a biodiversity concern feels weak. There is a nice discussion of the knowledge gap of the vertical structure of the water vapor field in the lower troposphere above Alpine lakes. It would feel more natural to tie the lake influence to the surface energy budget and atm-surface interactions in complex terrain, e.g. Wang, Wei, Xuhui Lee, Wei Xiao, Shoudong Liu, Natalie Schultz, Yongwei Wang, Mi Zhang, and Lei Zhao. "Global Lake Evaporation Accelerated by Changes in Surface Energy Allocation in a Warmer Climate." Nature Geoscience, 2018, 1–7. https://doi.org/10.1038/s41561-018-0114-8.

**The first paragraph has been removed to go directly to the moisture profiles as suggested by the editor.**

Line 340 – Switching interval between intakes? Or physically move the sampling inlet? Length of tubing, tubing type, flow rate?

**We used 2.5 m of 1/4" PTFE tubing and a 40cm 1/4" stainless steel tip on the first inlet, and about 1.5m of tubing with a 50 cm stainless steel tip. A flow rate of about 10 lpm was provided by an external manifold pump (N022AN, KNF, Germany) to either of the selected inlet lines. These details have been added to the revised manuscript.**

Are figures 7-9 all necessary?

**No, it was reduced to 1, but this was in response to the request of the other reviewer.**

Fig 14b: any way to make flight 10 (green) data visible below 3,000m. Discussed on line 490.

**To make the flights more visible, the figure has been divided in two. Flights with a gradient have been separated (Fig. 6b-c in the new version).**

Fig 14a: what are the values of the 2 mixing sources?

**The values of the end point have been added in the figure caption of Fig. 6.**

Fig 15a: are there no vapor observations on 20 Jun AM and 21 Jun?

**In the table of Appendix A there is data on 20 June, but not on 21 June.**
**In Table B2, there is data on 20 June with flights 14 and 15, but not on 21 June.**
**On 20 June AM there is data for flight 14 (late morning/midday).**
**All information is present.**

Fig 15b: I don't see an orange dot.

**The orange dot (sampling) is not available on 22 June. The correction has been done in the text.**

Fig 16: The artwork is lovely, but why not show actual RH profile data? Why does RH increase from 1 km to 2.5 km? But 10 km lake regional influence is within the lower 2.5 km?

**This figure has been removed.**

Line 531-547: I think the authors are referring to the lake-atm interface layer as the lake water surface layer. However, when discussing equilibrium vapor throughout, it's hard for the reader keep the liquid frame of reference.

**This section has been reorganized to make it clearer.**

536-547 could be cut or moved after the next paragraph perhaps?

**The correction has been made.**

Fig 15b: Lake at 2m depth -9.5, the values with very low dex at 2m are very curious. I would not have expected that other than the surface water samples.

**We don't understand the reviewer comment. There is no data point with a dex of -9.5 for the 2m samples. Maybe the reviewer looked at the circle points, which are microlayer? The only point with an unusually low dex at 2m is from 16 June (blue rhomb), with about -14 permil. This data point stands out with a low d-excess, but that we have no indication for evaporation loss after sampling. More systematic measurements would be needed to confirm if this is a realistic measurement value.**

579: confidence interval looks like an equation where the terms are multiplied, but that is not the intent. Maybe place a comma between them?

**The correction has been made.**

Fig 17: Add the statistical description of the box-whisker plot to the figure caption. The confidence intervals are so small compared to the 25th and 75th percentiles. Is that because n for the vapor measurements so large? What if you did 1-min averaging instead of 10 sec averaging of the data? That would decrease n by a factor of 6 and widen the confidence intervals without influencing the mean values. What is an accurate representation of statistically significant differences?

**What was once in the text has been added to the legend but removed from the text. Averaging does not change the median value. On the other hand, it will decrease the dispersion. In terms of the presentation of the calibrated data, we have chosen to keep the temporal resolution and therefore the vertical resolution. This is important because we make airborne measurements, and we need to sample quickly enough to observe atmospheric variability. Subsequently, the vertical profiles are degraded in vertical resolution in order to improve the signal-to-noise ratio of the data (new subsection 6.1) while keeping the main vertical structures.**

Line 595-597: this sentence is vague. Can you be more specific about what values are estimated for ET?

**This sentence may be too speculative and has been removed. Nevertheless, using the nearby IAEA measurements in precipitation, one can take the summer precipitation values as an upper bound of what would be contributed by vegetation (Based on data available until 2018, typical summer $\delta 18O$ ($\delta 2H$) values fall within the interval [-16, -1] ‰ ([-120, -20] ‰). In addition, there would be a contribution from other seasons to groundwater, drawing the vegetation signal to more negative values.**

Line 649-650: "Moreover, the $\delta O18$ in equilibrium condensate above the lake is generally substantially more depleted, confirming the existence of non-equilibrium fractionation during lake evaporation." This could just mean relatively little lake influence compared to advection.

**Indeed, the correction has been made.**

---

## Author Response (AR3)

Dear Editor,

At the request of the editor and the reviewer, the following corrections have been made:

   *- line 561: retrieved --> retrieve*

   *- line 576: "in isotopes abundance" --> "in the abundance of isotopes"*

   *- line 614: reference seems to be incomplete*

   *Line 481: I think the authors intended to say, 'the atmosphere near the lake shows near-surface stability over 200 to 300 m which deteriorates/mixes away in the afternoon according to the potential temperature profiles.' The rest of the paragraph contradicts the existing first sentence.*

   *Fig D2: please add 'ground (black) and flight (grey)' in the legend.*

Hoping to enlighten the reviewer on his next remark:

*In Fig 8, I would be interested to learn how the authors interpret the ASR vertical profiles in the context of the atmospheric stability and mixing. It seems to have to more or different structure compared to the other variables.*

Lidar measurement generally show many more vertical structures and have a higher vertical resolution. They are also more localized in time. What is most important to compare with in situ measurements are the major slope changes. By averaging the in-situ data, rapid variabilities are dampened. A good example is given in Fig. 9c. There is a change in abundance of $\delta D$ around 2 km a.m.s.l. This change is associated with a slight difference in the vertical gradient of potential temperature and is much more pronounced in the specific humidity. On the ASR (lidar) side, the change in vertical gradient is associated with a maximum that reflects the transition between the valley air and that influenced by the larger scale circulation above. As for the top of the atmospheric boundary layer, a stronger backscatter is observed, mainly associated with the increase in relative humidity which leads to the growing of aerosols. ASR is therefore a good tracer for the identification of vertical structures. On the other hand, it does not provide information on vertical thermal stability, which is more accurately traced by the potential temperature. Obviously, experience in using this type of measurements allows one to know if the layer is more or less stable. Generally speaking, the more the transition is peaked on the ASR, the more we are dealing with a convective layer. For example, there is less convection in Fig. 8a than in Fig. 8b, and this varies in the same way as the vertical stability. Nevertheless, we can have very convective and stable layers. For this reason, it is preferable to use different independent variables to conclude, the ASR being one of them.